# AUTOMATED ALGORITHM DESIGN WITH LLMS: A BENCHMARK-GUIDED APPROACH TO BLACK-BOX OPTIMIZATION

## ABSTRACT

Large Language Models (LLMs) have already been widely adopted for automated algorithm design, demonstrating strong abilities in generating and evolving algorithms across various fields. Existing work has largely focused on examining their effectiveness in solving specific problems, with search strategies primarily guided by adaptive prompt designs. In this paper, through investigating the token-wise attribution of the prompts to LLM-generated algorithmic codes, we show that providing high-quality algorithmic code examples can substantially improve the performance of the LLM-driven optimization. Building upon this insight, we propose leveraging prior benchmark algorithms to guide LLM-driven optimization and demonstrate superior performance on two black-box optimization benchmarks: the pseudo-Boolean optimization suite (pbo) and the black-box optimization suite (bbob). Our findings highlight the value of integrating benchmarking studies to enhance both efficiency and robustness of the LLM-driven black-box optimization methods. The source code and auxiliary materials are provided at `https://anonymous.4open.science/r/ICLR2026-submissionID2452-D709`.

## 1 INTRODUCTION

The emergence of Large Language Models (LLMs) has elevated automated algorithm design (AAD) to a new level, enabling the direct evolution of algorithms by optimizing code. In contrast, traditional AAD focused on algorithm configuration (Schede et al., 2022), relying on tuning parameter values and selecting operators. After the pioneering work of FunSearch (Romera-Paredes et al., 2024) demonstrating the potential of LLMs by solving the cap set (Grochow, 2019) and bin packing (Coffman Jr et al., 1984) problems, LLM-driven optimization methods have been applied to solve problems across specific domains and to evolve existing algorithmic frameworks. They have achieved significant success in diverse applications such as scheduling and routing, satisfiability, multi-objective optimization, black-box optimization, etc. (Liu et al., 2024a; Sun et al., 2025; Yao et al., 2025; Huang et al., 2025; van Stein & Bäck, 2024), as well as in advancing modularized algorithmic frameworks including Bayesian Optimization and Local Search for Pseudo Boolean Optimization (Liu et al., 2024b; van Stein et al., 2025; Li et al., 2025a;b). Moreover, the LLM-driven AAD methods evolved from relying on massive scale sampling of LLMs (on the order of $10^6$) to employing evolutionary approaches that require only hundreds of samples.

Search strategies are recognized as an essential component of LLM-driven optimization methods (Zhang et al., 2024), which are embedded with variation operators, such as generating diverse algorithms, that are accomplished through specific prompt design. However, these prompts have been largely designed based on intuition in existing work, with the assumption that LLMs would respond reliably to linguistic instructions. This fact raises concerns about the underlying behaviour of LLMs and hinders the development of more efficient, robust, and reliable LLM-driven approaches. To address this, we apply AttnLRP (Achtibat et al., 2024), an attention-aware feature attribution method, for the first time to investigate the token-wise contribution of prompts in code generation and LLM-driven optimization studies, aiming to uncover the mechanisms driving LLMs' behaviour and to design more effective and robust LLM-driven optimization approaches.

Our study focuses on black-box optimisation (BBO, defined in Appendix C), which does not provide an internal structure of the objective function. In practice, the algorithms can access only the problem metadata (e.g., variable domain, dimensionality) and the fitness values of explored solutions during the optimization process. However, existing LLM-driven approaches (van Stein & Bäck, 2024; Liu et al., 2024a) commonly embed prior knowledge, such as problem names, into their prompts. While this is a reasonable choice to effectively guide LLMs toward producing useful outcomes, it remains crucial to carefully consider practical constraints when developing tools for BBO. Meanwhile, traditional AAD for BBO relies on algorithm configuration and algorithm selection methods, which are based on self-guided search or feature-based learning techniques to construct competitive algorithms within predefined frameworks for specific tasks (Schede et al., 2022; Kerschke et al., 2019). Extensive benchmark studies have complemented these efforts, offering valuable guidelines for building and evaluating these techniques (Bartz-Beielstein et al., 2020; Bennet et al., 2021). In contrast to domains focusing on practical problems such as Satisfiability (SAT) and travelling salesman problem, where commonly accepted algorithms and benchmark rankings are available, selecting an appropriate algorithm for a given BBO remains challenging. Different BBO algorithms often obtain fundamentally different algorithmic structures, making it impractical to leverage LLMs to optimize particular algorithmic modules as in prior work (Liu et al., 2024a; Sun et al., 2025). Instead, in this paper, we leverage a set of benchmark algorithms to guide LLMs toward generating improved algorithms.

In this paper, we study LLM-driven optimization approaches by strictly adhering to the *black-box* settings, ensuring that no prior knowledge from tested suites is exposed. Furthermore, inspired by our investigation on token-wise contributions of prompts, we demonstrate that leveraging prior benchmark algorithms can effectively guide LLMs towards superior and more robust performance. With extensive experiments on the pseudo-boolean optimization (pbo) suite (Doerr et al., 2020) and the continuous black-box optimization (bbob) suite (Hansen et al., 2021), we demonstrate the advantages of integrating established benchmark practice with LLM-driven approaches. Our findings highlight that this integration will benefit not only BBO but also the broader field of LLM-driven optimization.

Overall, this work contributes to:

- A systematic analysis of the token-wise contribution of prompt design in the LLM-driven optimization frameworks, demonstrating that the embedded example codes obtain the most significant impact on the algorithmic codes produced by LLMs.

- An explicit demonstration that the behaviour of LLMs can be effectively guided through providing specific example codes, restricting the algorithmic search regions of LLMs.

- A competitive benchmark-guided approach that outperforms existing LLM-driven optimization methods on two well-established BBO benchmarks, *pbo* and *bbob*. The proposed benchmark-guided technique provides new insights into designing efficient, robust, and reliable LLM-driven optimization methodologies for future work.

## 2 RELATED WORK

### 2.1 LLM-DRIVEN OPTIMIZATION

Since the success of Funsearch (Romera-Paredes et al., 2024), which demonstrated the use of LLMs to solve the cap set and bin packing problems, LLM-driven optimization techniques have been widely applied in various fields. Evolution of Heuristics (EOH) (Liu et al., 2024a) addressed Traveling Salesman and Flow Shop Scheduling problems by leveraging LLMs to evolve algorithms and their own *thoughts* of producing new codes within a predefined algorithmic skeleton. In contrast, LLAMEA (van Stein & Bäck, 2024) applies LLMs to directly generate metaheuristics, achieving competitive performance on the continuous black-box optimization. Similar techniques of these works have also been applied for multi-objective optimization (Yao et al., 2025) and Bayesian optimization (Liu et al., 2024b; Li et al., 2025b). In SAT, the AutoSAT framework (Sun et al., 2025) pioneered optimizing SAT solvers with LLMs, followed by NVIDIA Research, which recently achieved significant improvement on the state-of-the-art performance (Yu et al., 2025). Meanwhile, more recent efforts have also targeted the domain of constrained PBO (Li et al., 2025a).

Unlike the pioneering FunSearch, which requires a significant number of LLM queries to evolve algorithmic codes, recent work adopts an evolutionary procedure loop, as illustrated in Figure. 1. This loop starts with an initial prompt design and iteratively queries LLMs for new algorithm variants with online refined prompts. In the context

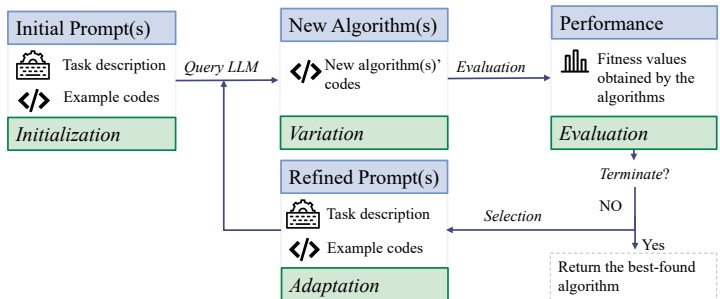

Figure 1: The workflow of LLM-driven optimization approaches

of evolutionary computation (Bäck et al., 2023), this process follows the stages of *initialization, variation, evaluation,* and *adaptation.* Notably, the *variation* and *adaptation* are driven by LLMs and heavily influenced by prompt design. A recent survey highlights the essential role of search strategies (Zhang et al., 2024), for example, the number of algorithms generated at each generation and the selection criteria directly affecting *adaptation.* Nevertheless, prompt design remains the most distinct aspect across applications of LLM-driven optimization, determining how effectively LLMs guide the search process.

## 2.2 ATTRIBUTING INPUT PROMPTS OF DECODER-ONLY LLMS

State-of-the-art LLMs typically allow up to hundreds of thousands of input tokens and operate in an excessively black-box manner. The long context window not only enables rich task descriptions from users, but also raises two questions: *Do we need all these input tokens? Do all input tokens contribute equally and positively?* To address these concerns, token-wise (feature) attribution methods are proposed to assist human understanding of the contributions of input tokens by providing comparable token-to-token relevance scores between an input prompt and its corresponding LLM's output. This type of explanation falls into local explanations and can be divided into four categories regarding their fundamental strategies (Zhao et al., 2024; Schneider, 2024): Perturbation-based methods (Li et al., 2016; Wu et al., 2020); Surrogate-based methods (Ribeiro et al., 2016; Kokalj et al., 2021); Gradient-based methods (Sundararajan et al., 2017; Enguehard, 2023), and decomposition-based (Montavon et al., 2019). The cutting-edge methods tailored for LLMs, including AttnLRP (Achtibat et al., 2024), Progressive Inference (Kariyappa et al., 2024), and JoPA (Chang et al., 2025), are essentially building upon multiple strategies. To deal with complex input-output pairs in LLM-driven optimization, we use AttnLRP as the explainer in this paper. It is fundamentally grounded in layer-wise relevance propagation (Montavon et al., 2019) and empowered by gradient information, producing sparse and faithful explanations with fast inference.

## 2.3 BENCHMARKING BLACK-BOX OPTIMIZATION

While researchers work on automated algorithm design to achieve competitive performance with minimal development cost, benchmarks are also extensively studied in BBO to better understand algorithms' behavior across different types of problems. One of the main goals is to provide comprehensive and fair comparisons of algorithms across diverse problem sets from multiple perspectives (e.g., considering multiple performance measures) (Bartz-Beielstein et al., 2020). The resulting benchmark data serve as valuable resources for the learning process in traditional automated approaches, such as algorithm configuration and algorithm selection (Schede et al., 2022; Kerschke et al., 2019). Several platforms have become commonly accepted in the BBO community. In practice, bbob and its variants provide well-established problem suites for continuous BBO, supported by the COCO platform (Hansen et al., 2021). More recently, a problem suite for pseudo-boolean BBO (pbo) has been proposed, accompanied by the IOHprofiler platform Doerr et al. (2020). Another platform, Nevergrad, integrates these problems and provides extensive algorithms as well as an automated algorithm selector (NGopt) based on problem meta information (Bennet et al., 2021). Furthermore, both problem suites, pbo and bbob, are embedded in a novel BBO benchmark framework Bencher (Papenmeier & Nardi, 2025).

## 3 BENCHMARKS

All experiments in this paper are evaluated on two benchmark suites that are widely used in the BBO community to ensure fair comparisons, pbo (Doerr et al., 2020) and bbob (Hansen et al., 2021). The pbo suite contains 23 pseudo-boolean optimization problems, while the bbob suite contains 24 continuous optimization problems in their standard settings. Both suites are well-established and have made significant contributions to algorithm development in their respective domains. In addition, these benchmark suites have attracted attention beyond BBO and are also widely applied in other domains such as general benchmark (Papenmeier & Nardi, 2025), learning (Ma et al., 2025), and algorithm configuration (Ye et al., 2022; Li et al., 2024; Song et al., 2025). More details of the benchmark problems are provided in Appendix C.

Performance measures are a crucial aspect in benchmarking and analyzing the behavior of algorithms. Unlike traditional indicators, considering the best-found solution within a fixed time or the time required to reach a specific target, anytime performance measures have recently been widely accepted in the BBO community (Hansen et al., 2022; Wang et al., 2022). In this work, we evaluate all BBO algorithms using the (approximated) area under the ECDF curve (AUC). Given a set of time points $T$ and a set of target values $\Phi$, the ECDF value of an algorithm at time $t \in T$ is defined as the fraction of target values in $\Phi$ that are worse than the best-found fitness obtained by $A$ up to time $t$. The AUC value of $A$ is then computed as the aggregation of ECDF values across all $t \in T$. In this paper, we normalize it by the size of $T$. The specific settings of $T$ and $\Phi$ used in our experiments are reported in Section 6 and Appendix C. Throughout this work, we evaluate algorithm performance on a problem by averaging the AUC over five problem instances (see Hansen et al. (2010); Doerr et al. (2020)) to ensure robustness.

## 4 TOKEN-WISE ANALYSIS OF PROMPT DESIGN IN LLM-DRIVEN BBO

In the evolutionary procedure of searching for improved algorithms within LLM-driven optimization approaches, querying LLMs for new algorithms plays an essential role in the *variation* step, as illustrated in Figure 1. Designing more effective search strategies for LLM-driven approaches, therefore, requires a deeper understanding of how LLM produces new algorithms. Such insights are not only valuable for explaining the evolutionary procedure but also for enabling precise control over the search process, eventually leading to better and robust results. For example, restart strategies and diversity control are two common techniques in evolutionary computation, highlighting this need for proper technique design. Restart strategies help escape from local optima by reinitializing the search state, while diversity control ensures the exploration of sufficiently different solutions to prevent premature convergence. Current attempts at incorporating these advanced techniques rely on prompt engineering of *linguistic task descriptions*, such as *"creating a novel algorithm"* or *"exploring new heuristics"* (Liu et al., 2024a; van Stein & Bäck, 2024). However, a systematic understanding of how prompts influence code generation in LLM-driven optimization remains limited.

To address this gap, we investigate in this section the research question *Q1: How does the prompt design affect the generated algorithmic code?*, by analyzing the token-wise contribution of the prompt to the algorithmic codes generated by LLMs.

Specifically, we leverage AttnLRP (Achtibat et al., 2024), an attention-aware feature attribution method that adapts the classic Layer-wise relevance propagation (LRP) (Montavon et al., 2019), to identify the contribution of input prompt tokens to the generated algorithms. The classic LPR assumes that a function $f_j$ with input $N$ features $\boldsymbol{x} = x_i, i \in \{1, \ldots, N\}$ can be decomposed into individual contributions of each feature $R_{i \leftarrow j}$, which is called *relevances*. $R_{i \leftarrow j}$ quantifies how much of the output $j$ is attributed to input feature (token) $i$, and LRP calculate the relevance of feature $i$ by summing across all outputs, $R_i = \sum_j R_{i \leftarrow j}$. By treating a neural network as a layered directed acyclic graph, LRP denotes a neuron $j$ in layer $l + 1$ as a function node $f_j^{l+1}$. It enforces a *conservation property*, such that relevance score are redistributed backwards layer by layer while preserving their total values:

$$R^l = \sum_i R_i^l = \sum_{i,j} R_{i \leftarrow j}^{(l,l+1)} = \sum_j R_j^l = R^{l+1}$$

Following this principle, we can compute the relevance of each token $R^0$ by starting from the model's output and propagating all the way back to the input. We adopt the decomposition rules of $R_{i \leftarrow j}$ proposed in AttnLRP, with details provided in Appendix B.

Our analysis focuses on the contribution of the input prompt tokens to the output of querying decoder-only LLMs to generate novel algorithms. For each output token $j \in T_{out}$, we compute the relevance $R_i^0$ for the input tokens $i \in T_{in}$ and a set of previously generated output $j \in \mathcal{P}(T_{out})$, since LLMs apply autoregressive models, generating each token conditioned on previously generated tokens and the input. Thus, for each output $j$, we obtain relevance $R_i^0, i \in T_{in} \cup \mathcal{P}(T_{out})$. To fairly compare the contributions of input tokens with respect to a set of output tokens (e.g., a code segment), we truncate and aggregate the normalized values into $\tilde{R}_i^0, i \in T_{in}$.

Table 1: Mean and standard deviation of relevance scores ($\times 10^{-3}$) across different parts of prompts. The scores are normalized values whose sum equals 1.

| Results on instruction-tuned 27b Gemma 3 | | | | | | |
|---|---|---|---|---|---|---|
| Task Description | Strategy | Expected Output | Note | Parent Heuristic(s) | Code description | Linguistic description |
| $0.983 \pm 0.192$ | $2.133 \pm 0.062$ | $1.082 \pm 0.163$ | $0.340 \pm 0.066$ | $1.912 \pm 0.328$ | $\mathbf{2.194 \pm 0.238}$ | $0.807 \pm 0.029$ |
| Results on instruction-tuned 14b Qwen 2.5 Coder | | | | | | |
| Task Description | Strategy | Expected Output | Note | Parent Heuristic(s) | Code description | Linguistic description |
| $0.864 \pm 0.168$ | $1.383 \pm 0.275$ | $0.898 \pm 0.175$ | $0.302 \pm 0.048$ | $1.279 \pm 0.108$ | $\mathbf{1.399 \pm 0.141}$ | $0.656 \pm 0.054$ |

**Experiments** We compute token-wise contributions for pairs of prompts and outputs in an elitist evolutionary procedure, where LLMs generate a new algorithm code at each iteration. If a generated algorithm outperforms the current best, we adapt the prompt by using its code. Specifically, we leverage AttnLRP to explain the outputs of two open-source LLMs, the instruction-tuned 27B Gemma 3 and 14B Qwen 2.5 Coder, for automatically designing algorithms to solve the classic OneMax problem (F1 in pbo, see details in Appendix C). Figure 2 presents a heatmap of the prompt relevance to the corresponding generated algorithmic code, clearly showing that the code-related content obtains a strong influence. Additional heatmaps of prompt-code pairs in our public repository also demonstrate consistent behaviour.

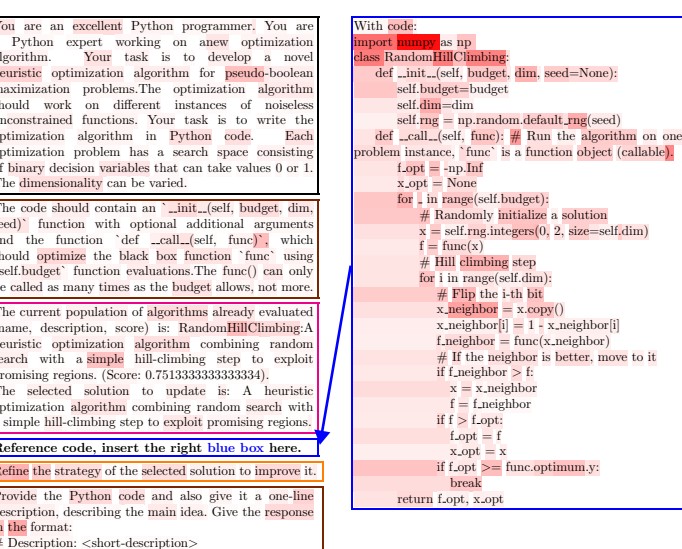

Figure 2: The heatmap of the token-wise relevance of a given prompt to its corresponding newly generated algorithmic code. The result is obtained on an instruction-tuned 27b Gemma 3 LLM using the AttnLRP explainer. Darker shading indicates higher aggregated relevance scores, thus more important.

To quantify the contributions of different prompt elements, we partition prompts into five components: *Task Description, Strategy, Expected Output, Note, and Parent Heuristics*, following the setting of prior work (Liu et al., 2024a). Table 1 reports the average tokens *relevance* of each component across all prompt-code pairs for the first function (OneMax) in the PBO set, as generated by the instruction-tuned 27B Gemma 3 model. For *Parent Heuristic(s)*, which consists of both code and linguistic description, we further separate it into two parts. *Strategy* is defined as a short instruction indicating whether to *refine* the provided code or *create* a novel algorithm. *Note* specifically

contains the fitness value (e.g., score) obtained by the parent code. Additional details are explained in Appendix D.

The results demonstrate that the *code description* and the associated *strategy* obtain the strongest influence on the produced algorithmic codes, consistently across both tested LLMs. Meanwhile, the linguistic descriptions of tasks and the current algorithms contribute less to the overall code-related content. In particular, the fitness value of the parent code, which is important for the *selection* of evolutionary procedures, does not exhibit a significant impact on the behavior of LLMs.

Overall, our extensive experiments demonstrate that *among all prompt components, the provided code example and its associated strategy instruction obtain the most significant impact on the output algorithms generated by LLMs*

## 5 GUIDING LLMS TOWARD SPECIFIC SEARCH REGION

Motivated by the significant influence of code-related content, we investigate in this section the research question: *Q2: Can we control LLMs to explore specific regions of algorithms?*

To address this question, we compare algorithms produced when LLMs are prompted with different example codes. Each run begins with a distinct example code, and we iteratively query LLMs to refine the current best code following the elitist strategy (see the prompt in Appendix D). Our hypothesis is that, by providing specific example codes, using LLMs to refine algorithmic code will behave as a neighborhood search, constraining exploration to a local region of the search space. It is worth mentioning that our focus lies in improving the LLM-driven optimization process itself while controlling the evolutionary search towards better algorithms more efficiently. Specifically, we construct prompts that integrate example codes to guide LLMs towards generating promising, diverse algorithm candidates. This differs from the concept of *in-context learning* (Dong et al., 2024), which focuses on adapting LLMs' behaviour with a single prompt rather than exploring the search space of algorithms through optimization techniques.

We compare this refinement strategy with EoH (Liu et al., 2024a), LLaMEA (van Stein & Bäck, 2024), and ReEvo (Ye et al., 2024). These three state-of-the-art approaches leverage stochastic mechanisms that enable different strategies, such as modifying the given heuristics, creating an entirely new algorithm, and other variators. In contrast, we test a refinement strategy (Refine:A$i$) that iteratively queries LLMs to *refine* the current best algorithmic code, starting with a given example code A$i$. Experiments are tested on pbo problems, and we test different settings by using the top five algorithms A$i$, $i \in \{1, \ldots, 5\}$ for each problem based on the benchmark data in Doerr et al. (2020).

Figure 3 presents the results for OneMax (F1 of pbo, see Appendix C). Additional experimental results are available in Appendix G. We can observe that Refine:A$i$ methods yield distinct performance trajectories, which confirms that different example codes can steer LLMs towards different search regions. Meanwhile, refining certain algorithms can outperform the state-of-the-art baselines, achieving the best AUC across three tested LLMs. However, identifying the suitable example code that can guide LLMs toward the optimal performance remains an open question.

Overall, the results indicate that *embedding specific example code can constrain the search region of LLMs*, while raising the question of how to identify and effectively utilize such codes.

## 6 A BENCHMARK-GUIDED APPROACH

To address the research question *Q3: how can we effectively control the search process of LLM-driven optimization approaches?*, we propose a benchmark-assisted guided evolutionary approach (BAG) following the evolutionary procedure as described in Algorithm 1. Specifically, the prompt $P(\mathcal{A})$ used to query the LLM is consistently embedded with an example code of the current algorithm $\mathcal{A}$. A benchmark set of algorithms, denoted as $\mathbf{A}_{bench}$, is leveraged to guide the prompt design during both the *Initialization* and *Adaptation* steps. Recall that determining the optimal algorithms for an arbitrary BBO problem is challenging, and the conclusion may vary considering different performance measures. Therefore, to avoid being trapped in a particular algorithmic pattern, we work with a set of benchmarks, $\mathbf{A}_{bench}$, rather than relying on a single algorithm. Note that the algorithm refers to its code in the following description. The BAG method is detailed as follows.

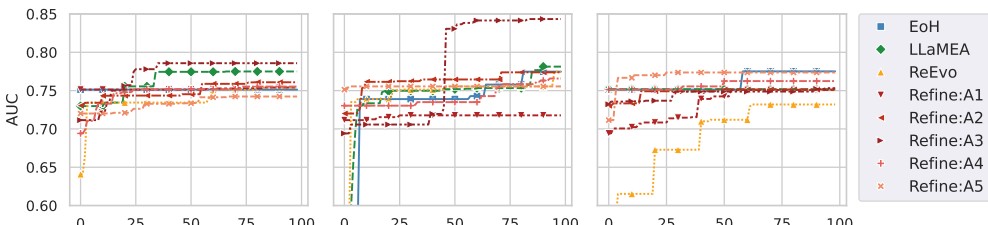

Figure 3: Convergence process of LLM-driven approaches searching algorithms for OneMax. $x$-axis represents the number of algorithms generated by the LLM, and $y$-axis indicates the best-so-far AUC value. Results are from using Gemini, GPT, and Qwen, respectively (from Left to Right).

- We adopt the (1+1) elitist search strategy, which generates a new algorithm at each iteration. Its effectiveness has been demonstrated by theoretical and empirical results (Witt, 2006; Droste et al., 2002; Hoos & Stützle, 2018) on mutation-based evolutionary algorithms and local search. These methods are analogous to our motivation of guiding LLMs to explore the neighborhood of the provided code example.

- *Initialization* is performed using a promising example code selected from $\mathbf{A}_{bench}$ (line 2 in Algorithm 1), rather than through random sampling.

- At each iteration, the LLM is queried either to *refine* the current best algorithm $\mathcal{A}^*$ or to *create* a novel algorithm with equal probability (lines 11-14), suggested by prior LLM-driven approaches (Liu et al., 2024a; Ye et al., 2024; van Stein & Bäck, 2024). The *refine* operator incrementally optimizes $\mathcal{A}^*$ under the elitist strategy, while the *create* operator introduces diversity, though it heavily relies on the generative capacity of LLMs.

- To further exploit benchmark knowledge, every $q$ iterations the LLM is specifically queried to *refine* an algorithm randomly selected from $\mathbf{A}_{bench}$ (lines 8-9). This encourages exploration of promising regions identified in prior benchmark while maintaining diversity.

The benchmark algorithm selection follows a sampling without replacement. Once all elements of $\mathbf{A}_{bench}$ have been selected, the process restarts with the full set $\mathbf{A}_{bench}$. $rand \in [0, 1]$ denotes a value sampled uniformly at random. If a generated algorithm $\mathcal{A}$ fails to execute or does not return a solution within the timeout, its fitness $F(\mathcal{A})$ is assigned the worst possible value. We set $q = 10$ as suggested by the experimental analysis in Appendix F.

**Technique differences from existing methods**    The key innovation of our proposed BAG framework lies in leveraging prior benchmark knowledge to effectively guide LLMs toward specific regions and discover improved algorithms. To this end, BAG applies a (1+1) mutation-based elitist scheme, chosen to clearly highlight this motivation. By contrast, EoH and ReEvo were originally introduced with population-based schemes that incorporate both mutation and crossover-like operators on algorithmic code, and LLaMEA exhibits both the (1+1) and population-based strategies. Furthermore, existing approaches primarily adapt the *Strategy* component of prompts to control the search process. For example, EoH is motivated by the idea that *the evolution of thought, a linguistic description representing a high-level idea of a heuristic, is important* (Liu et al., 2024a). However, BAG emphasizes adapting *code* itself as the core of prompt design and search guidance, while relying on only two concrete strategies, refining the current algorithm or generating a novel one.

**Experimental settings**    We compare BAG with EoH, LLaMEA, and ReEvo across $47 \times 5$ problem instances, including 23 100-dimensional pbo problems and 24 5-dimensional bbob problems. The details of the benchmarks are described in Appendix C. For the construction of $\mathbf{A}_{bench}$, we select the top five algorithms reported in the repository of (Doerr et al., 2020) for each pbo problem. Since no repository of standard formatted code that supports effective LLM interaction for bbob, we form $\mathbf{A}_{bench}$ using five widely adopted algorithms in continuous BBO: covariance matrix adaptation evolution strategy (CMA-ES) (Hansen, 2016), Cholesky CMA-ES (Krause et al., 2016), evolution strategy with cumulative stepsize adaptation (Chotard et al., 2012), differential evolution (Das & Suganthan, 2010), and particle swarm optimization (Wang et al., 2018).

**Algorithm 1:** A Benchmark-assisted LLM-driven BBO

---

1 **Input:** A set of or a problem and a fitness measure $F$, a set of benchmark algorithms $\mathbf{A}_{bench}$, a prompt template $P$, a LLM model, a frequency factor $q$, and a maximal budget $\mathcal{B}$ of querying the LLM;

2 **Initialization:** Select a preferred benchmark algorithm code $\mathcal{A}^*$ from $\mathbf{A}_{bench}$, generate an algorithm code $\mathcal{A}$ by querying the LLM with the template prompt $P(\mathcal{A}^*)$ (***Initialization***);

3 Evaluate the algorithm by $F(\mathcal{A})$, $t \leftarrow 1$;

4 **if** $F(\mathcal{A})$ *outperforms* $F(\mathcal{A}^*)$ **then** $\mathcal{A}^* \leftarrow \mathcal{A}$;

5 **while** $t < \mathcal{B}$ **do**

6     (***Adaptation & Variation***);

7     **if** $t \mod q = 0$ **then**

8         Select an algorithm randomly from $\mathcal{A}' \in \mathbf{A}_{bench}$;

9         Generate an algorithm $\mathcal{A}$ by querying the LLM with the prompt $P(\mathcal{A}')$ to *refine* $\mathcal{A}'$;

10     **else**

11         **if** $rand < 0.5$ **then**

12             Generate an algorithm $\mathcal{A}$ by querying the LLM with the prompt $P(\mathcal{A}^*)$ to *refine* $\mathcal{A}^*$;

13         **else**

14             Generate an algorithm $\mathcal{A}$ by querying the LLM with the prompt $P(\mathcal{A}^*)$ to *create a novel algorithm*;

15     Evaluate the generated algorithm by $F(\mathcal{A})$, $t \leftarrow t + 1$ (***Evaluation***);

16 **if** $F(\mathcal{A})$ *outperforms* $F(\mathcal{A}^*)$ **then** $\mathcal{A}^* \leftarrow \mathcal{A}$;

17 **Output:** Return the best algorithm $\mathcal{A}^*$.

---

We apply the default configurations of EoH, LLaMEA, and ReEvo as specified in their publications. Experiments are conducted with three LLMs: Google's Gemini 2.0 Flash (Gemini), OpenAI's GPT 5 Nano (GPT), and Alibaba's Qwen3 Coder Flash (Qwen) (see Appendix E for details). The budget of LLM query is limited to 100, and the cutoff time (i.e., maximal function evaluations) for evaluating algorithms on each problem instance is set to $10^4$ and $10^6$ for pbo and bbob, respectively, consistent with common practice for these two benchmarks (Doerr et al., 2020; Hansen et al., 2021). The performance of algorithms (i.e., fitness $F$) is evaluated by AUC (see Appendix 3), $T$ is formed by 100 log-scaled samples from 0 to the cutoff time, and the target set is introduced in Appendix C. Since this measure is defined in terms of function evaluations, the results reported in this paper are independent of the underlying hardware. Additional information can be found in Appendix E.

Table 2: Normalized best-achieved AUC results on 23 pbo problems and 24 bbob problems. We report the mean $\pm$ standard deviation (average rank). The best-performing entries are bolded.

| | Results on 23 pbo benchmark problems | | | |
|---|---|---|---|---|
| LLM | BAG | EoH | LLaMEA | ReEVO |
| Gemini | $\mathbf{0.9564}_{\pm\mathbf{0.0652}}(\mathbf{2.0})$ | $0.9543_{\pm0.0578}(2.67)$ | $0.9398_{\pm0.0938}(2.38)$ | $0.9239_{\pm0.1169}(2.83)$ |
| GPT | $0.9535_{\pm0.0638}(2.42)$ | $\mathbf{0.9589}_{\pm\mathbf{0.0537}}(\mathbf{2.29})$ | $0.9298_{\pm0.0854}(2.62)$ | $0.9264_{\pm0.1024}(2.67)$ |
| Qwen | $\mathbf{0.9751}_{\pm\mathbf{0.0393}}(\mathbf{1.88})$ | $0.9697_{\pm0.0414}(2.08)$ | $0.947_{\pm0.0556}(2.71)$ | $0.9037_{\pm0.0718}(3.33)$ |
| | Results on 24 bbob benchmark problems | | | |
| LLM | BAG | EoH | LLaMEA | ReEVO |
| Gemini | $\mathbf{0.9345}_{\pm\mathbf{0.0942}}(\mathbf{2.12})$ | $0.8783_{\pm0.1513}(2.32)$ | $0.8758_{\pm0.1626}(2.44)$ | $0.8569_{\pm0.1684}(3.12)$ |
| GPT | $\mathbf{0.9337}_{\pm\mathbf{0.1692}}(\mathbf{1.64})$ | $0.6467_{\pm0.2816}(2.92)$ | $0.7663_{\pm0.2789}(2.28)$ | $0.5776_{\pm0.2935}(3.16)$ |
| Qwen | $\mathbf{0.9618}_{\pm\mathbf{0.0744}}(\mathbf{1.64})$ | $0.7894_{\pm0.2002}(2.56)$ | $0.7810_{\pm0.1895}(2.96)$ | $0.7371_{\pm0.2568}(2.84)$ |

**Performance** Table 2 presents the average normalized AUC and ranks of BAG and the compared LLM-driven approaches across the pbo and bbob suites, and Figure 4 displays boxplots of normalized AUC distributions. Normalization is performed as $\text{AUC}/\text{AUC}_{\text{best}}$, such that the best approach

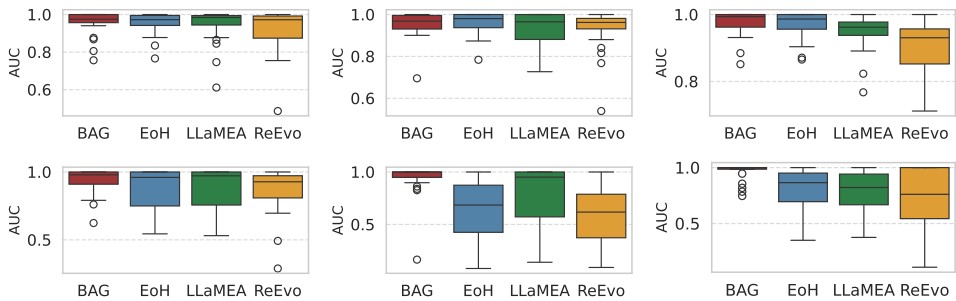

Figure 4: Boxplots of the best normalized AUC values obtained by the four approaches on 23 problems of pbo (Top) and 24 problems of bbob (Bottom). The LLM-driven approaches have been tested on Gemini, GPT, and Qwen, respectively (from Left to Right).

obtains a value of 1. Detailed results for each problem, including AUC values and the convergence trajectories, are provided in Appendix I.

*Overall Performance.* BAG consistently achieves superior performance. Specifically, it outperforms all baselines when using Gemini and Qwen for pbo, while it ranks second when using GPT, showing only a $-0.5\%$ gap in average AUC compared to EoH. Importantly, BAG requires only a single LLM query to obtain a novel algorithm candidate before evaluation, whereas EoH relies on multiple (five) prompts to generate an algorithm candidate. Given our budget for LLM-driven approaches is limited by the number of evaluations, BAG obtains the potential to achieve even better results under the same LLM query consumption as EoH. For bbob, BAG demonstrates significant advantages over the baselines, achieving on average a $14\%$ improvement over the second-best approach across all three tested LLMs. The competitive performance of BAG confirms

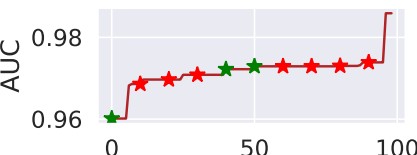

Figure 5: An illustration of the contribution of queries *refining* prior benchmark algorithms in BAG. The corresponding obtained results are marked by stars, and green indicates triggering improvements. Results are from using Gemini.

that benchmark knowledge can effectively guide LLMs toward promising regions of search space. Using prior benchmark algorithm code, BAG can initialize from a promising candidate (as presented in Figure 10-15 in Appendix I), ensuring final result quality and accelerating improvement.

*Convergence Analysis.* Figure 5 illustrates the convergence process of BGA on Sphere (F1 of bbob), where the star markers denote the fitness obtained from *refining* a new benchmark algorithm at fixed intervals (line 9 in Algorithm 1). The three green markers indicate triggered improvements, evidencing the effectiveness of our design in BAG. Apart from the convergence analysis in Figure 5, we also examine the long-term impact of prompt design by evaluating the similarity among generated algorithm codes using the CodeBLEU (Ren et al., 2020) metric (see Appendix H). Figure 6 presents the CodeBLEU scores for all ordered pairs of algorithms generated within the same evolutionary search procedure shown in Figure 5. We observe that the 41-st generated algorithm exhibits low similarity scores to all previously generated algorithms, as it is produced by introducing a new algorithm in $\mathbf{A}_{bench}$ for prompt design. However, subsequent algorithms show high similarity to this 41-st algorithm while obtaining low similarity to the first 40 ones. A Similar pattern can be observed for the 51-th generated algorithm, which is also generated by introducing new example code. These results indicate that the benchmark algorithms we incorporate play a crucial role in guiding the evolutionary search and substantially contribute to the superior performance of our BAG method.

*Additional Remarks.* The experimental results presented above are examined on 235 problem instances ($47 \times 5$), demonstrating the generalizability of our BAG method. We further assess the final obtained algorithms on five unseen instances for each problem, resulting in a total of 470 instances across training and testing. The corresponding results, which show consistent relative performance, are provided in Appendix J. In addition, since LLMs may generate code that fails to execute during the search process, we report the frequency of such situations in Appendix K.

Despite its simple search strategy design, BAG outperforms the state-of-the-art LLM-driven approaches, demonstrating potential for improvements with benchmark knowledge. For instance, the five benchmark algorithms used in our experiments are drawn from a limited dataset for each problem suite, while improvements could be expected by incorporating tailored codes, as discussed in Section 5. Therefore, it would be valuable for future work to explore integrating LLM-driven optimization with benchmark studies such as Nevergrad (Bennet et al., 2021), which provides extensive algorithm collections but uses complex frameworks that hinder effective LLM interaction.

In summary, BAG demonstrates that a benchmark-guided search strategy can substantially outperform existing LLM-driven optimization methods. The experimental results highlight that *fusing benchmark data can enhance the efficiency and robustness of LLM-driven BBO approaches.*

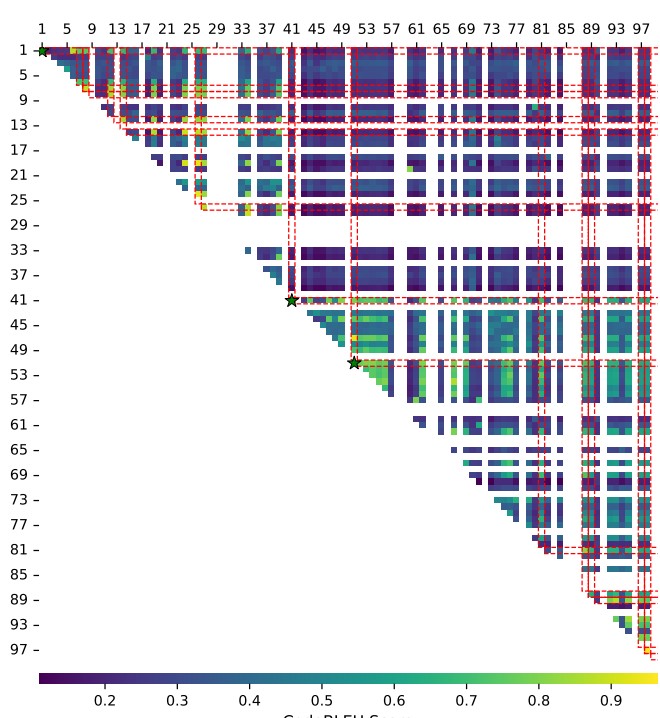

Figure 6: CodeBLEU similarity scores between ordered pairs of generated algorithms. Axes present time (i.e., the number of algorithms generated so far). Each column shows the relevance scores between a newly generated algorithm and all previously generated ones. Larger scores indicate higher relevance.

## 7 CONCLUSIONS

In this paper, we addressed the challenges of *understanding the impact of prompt design* and *controlling the search process in LLM-driven optimization.* We conducted the first relevance analysis of prompt components in code generation of LLM-driven optimization approaches by using AttnLRP, confirming that the code-related content obtains the strongest influence. Motivated by this observation, we propose the benchmark-assisted guided evolutionary approach (BAG). By combining prior benchmark knowledge with a simple yet effective (1+1) elitist search strategy, BAG provides a principal way of guiding LLMs towards promising search regions of algorithm codes.

Extensive experiments on two BBO suites demonstrated the superior performance of BAG, compared to three state-of-the-art LLM-driven optimization approaches, EoH, LLaMEA, and ReEvo. These results were consistent across 235 problem instances using three advanced LLMs.

These findings confirm that benchmark knowledge can effectively guide LLM-driven optimization, offering both practical improvements and a novel perspective on prompt design in LLM-driven black-box optimization. While BAG has achieved competitive performance under the (1+1) elitist scheme, EoH shows advantages in particular scenarios, benefiting from the population-based design. A natural next step is to integrate benchmark knowledge into a population-based framework, which can enhance both the performance and robustness of LLM-driven approaches by enabling more diverse and efficient search dynamics. Furthermore, we will conduct research on the self-learning scheme to advance both the fundamental benchmark and LLM-driven BBO.

## 8 REPRODUCIBILITY STATEMENT

The source code and auxiliary materials are anonymously available at `https://anonymous.4open.science/r/ICLR2026-submissionID2452-D709`. The repository includes clear instructions for reproducing all experiments described in this paper, along with the raw data and logs used to generate the reported figures and tables. Randomness in our experiments is exclusively introduced by the sampling strategy of the proprietary LLMs, which is beyond the authors' control. All other procedures, including the choice of refinement algorithms within the BAG optimization loop, are fully reproducible by fixing the random seed. Additionally, we describe the experimental setups, along with details on the baselines and benchmark suite, in both the main text and Appendix E.

## 9 ETHICS STATEMENT

We fully acknowledge and will faithfully adhere to the Code of Ethics of ICLR 2026. After carefully reading the ICLR 2026 Code of Ethics, we hereby provide the following clarifications. This work does not involve human subjects, personal data, or sensitive information. All experiments are conducted on publicly available benchmark problem suites (pbo, bbob), which have no privacy or security concerns. The source code, data, and logs are released with clear instructions to ensure transparency and reproducibility. Our methods rely on both open-source and proprietary large language models. We acknowledge that these models may have biases from their pre-trained data. However, our use is restricted to black-box optimization. It is very unlikely that such biases can result in direct human impact. Nevertheless, we are aware of this limitation and encourage caution if similar techniques are applied in socially sensitive domains. The contributions are intended for scientific exploration in black-box optimization. We do not see immediate risks of harm or misuse from this research. As with all LLM-based methods, deployment in safety-critical applications should be preceded by additional safeguards and domain-specific validation. The authors declare no conflicts of interest or external sponsorship that could bias this work.

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

## A    STATEMENTS ON THE USAGE OF LARGE LANGUAGE MODELS (LLMS)

This research focuses on LLM-driven optimization methods, and thus LLMs are an integral subject of study. Detailed descriptions of how the models are applied are provided in the main text, and the five LLMs used in our experiments are listed in Appendix E. In addition, ChatGPT, a commercial LLM assistant, was used solely to polish and improve the linguistic quality of this manuscript. LLMs were not involved in the conceptualization, design, or ideation of this work.

## B    A WALK-THROUGH OF THE ATTNLRP METHOD

In this section, we review the techniques introduced in AttnLRP (Achtibat et al., 2024), a post-hoc explanation method for transformer-based models. This algorithm explicitly requires access to the model's computation graph and performs backward attribution in a manner similar to backprop-agation during training, but with different propagation rules. Starting from the next-token logit, the method propagates relevance scores (rather than gradients) backward through the stacked trans-former layers, applying specialized rules to each submodule (e.g., linear layers, attention layers, normalization layers). Notably, the total relevance scores remain consistent at every step. In the following, we provide a concise overview of the proposed rules for key modules, i.e., feed-forward networks, normalization layers, and attention modules. We refer the interested readers to the original article for full technical details.

**Feed-Forward Networks (FFN)**    *Linear layers* are handled with the $\epsilon$-rule (Bach et al., 2015). The intuition is that each input (neuron or feature) receives the relevance score from the previous layer in proportion to how much it contributed (the weighted input) to the neuron's activation, with a $\epsilon$ value to ensure mathematical stability. Identity rule (directly pass through) is applied to *element-wise non-linearities*, saying the non-linear activation functions. Relevance scores of outputs are assigned directly to corresponding input features.

**Normalization Layers**    The authors claim that directly applying the linearization technique, such as the $\epsilon$-rule, to normalization layers (e.g., RMSNorm) causes the vanishing of almost all relevance scores into the bias. To handle this issue, they patch these layers by treating the denominator in normalization as a constant and hence reducing the calculation into element-wise linear operations. Thus, the rules designed for FFN can again be safely applied.

**Attention Mechanism**    The attention mechanism (Vaswani et al., 2017) consists of bilinear matrix multiplications and a nonlinear softmax activation function. The authors mathematically prove that both operations introduce challenges for standard LRP methods, resulting in unfaithful explanations. Particularly, the bilinear multiplication makes it unclear how to fairly distribute relevance between the query and the key matrices. AttnLRP addresses this issue by splitting the relevance equally (50/50) between the two matrices. This division is justified through both Shapley and deep Taylor decompositions, which both show uniform contributions of input matrices. The issue of softmax comes with the normalization: boosting one logit necessarily suppresses the others. To correct this, AttnLRP assigns each input relevance in proportion to its contribution to its own output, but subtracts the portion that comes from the normalization.

## C BENCHMARK SETTINGS

### C.1 BLACK-BOX OPTIMIZATION

We first define the general task of (single-objective) Black-box Optimization (BBO). Let $f : \mathcal{X} \to \mathbb{R}$ be an objective function defined over a search space $\mathcal{X} \subseteq \mathbb{R}^n$, where $n$ denotes the dimensionality. In BBO, the explicit form, derivatives, or structural properties of $f$ are unknown. The optimization algorithm can only query $f$ through evaluations of the type

$$y = f(x), \quad x \in \mathcal{X},$$

where $y$ is commonly mentioned as objective value or fitness value. The optimizer seeks to find

$$x^\star \in \arg\min_{x \in \mathcal{X}} f(x) \quad \text{or} \quad x^\star \in \arg\max_{x \in \mathcal{X}} f(x).$$

A *black-box optimization algorithm* is thus an iterative procedure that generates a sequence $\{x_t\}_{t=1}^T \subseteq \mathcal{X}$ based solely on past queries $\{(x_i, f(x_i))\}_{i=1}^{t-1}$, without requiring analytical knowledge of $f$.

### C.2 THE BENCHMARK SUITES

We provide in this section the detailed problem list of the pbo and bbob suites, including the corresponding target sets used to calculate AUC values.

The pbo suite covers a wide range of discrete problems, including the theory-oriented OneMax, LeadingOnes, and their variants, as well as practical problems. The pbo problems list is as follows:

- **F1**: OneMax maximizes the number of one-bits in a bitstring. For the 100-dimensional problem, we use the target set $\{50, \ldots, 100\}$ to calculate AUC in this paper.
- **F2**: LeadingOnes maximizes the number of consecutive one-bits from the start of a bitstring until the first zero-bit. We use the target set $\{0, \ldots, 100\}$.
- **F3**: Harmonic maximizes the weighted sum of one-bits, where the weight of each bit is $i, i \in \{1, \ldots, n\}$. The target set is $\{2525 + 5i \mid i \in \{0, 505\}$.
- **F4-F10**: OneMax variants with dummy, neutrality, and epistasis transformations. The target sets are $\{25, \ldots, 50\}$, $\{45, \ldots, 90\}$, $\{11, \ldots, 33\}$, $\{50, \ldots, 100\}$, $\{20, \ldots, 51\}$, $\{0, \ldots, 100\}$, and $\{0, \ldots, 100\}$, respectively.
- **F11-F17**: LeadingOnes variants with dummy, neutrality, and epistasis transformations. The target sets are $\{0, \ldots, 50\}$, $\{0, \ldots, 90\}$, $\{0, \ldots, 33\}$, $\{0, \ldots, 100\}$, $\{0, \ldots, 51\}$, $\{0, \ldots, 100\}$, and $\{0, \ldots, 100\}$, respectively.
- **F18**: LABS (Low Autocorrelation Binary Sequences). The target set is $\{0.5 + 0.1i \mid i \in \{0, 450\}$.
- **F19-F21**: Ising models maximize the energy of a lattice model, considering the one, two, and three-dimensional instances, respectively. The target sets are $\{50, \ldots, 100\}$, $\{100, \ldots, 200\}$, and $\{150, \ldots, 300\}$, respectively.
- **F22**: MIVS (Maximum Independent Vertex Set). The target set is $\{-1, \ldots, 51\}$.
- **F23**: N-Queens, in pbo, the 100-dimensional problem corresponds to 10-Queens. The target set is $\{-2, \ldots, 10\}$.

We refer to the detailed definitions of problems to Doerr et al. (2020). The target sets are determined based on the corresponding benchmark data. To prevent confusion in our discussion, we provide the definition of OneMax (F1) here.

$$f_{\text{OneMax}} : \{0, 1\}^n \to \{0, \ldots, n\}, x \mapsto \sum_{i=1}^n x_i$$

The bbob suite consists of five categories of continuous problems. When calculating the ECDFs of algorithms, we commonly consider the objective domain $[10^{-8}, 100]$ and compute the *fraction* (see Section 3 by the distance to $10^{-8}$, following the common setup of using bbob. The list of bbob problems is as follows:

- **F1-F5**: Separable functions including: Sphere, Separable Ellipsoidal, Rastrigin, Büche-Rastrigin, and Linear Slope

- **F6-F9**: Functions with low or moderate conditioning: Attractive Sector, Step Ellipsoidal, original Rosenbrock, and rotated Rosenbrock.

- **F10-F14**: Functions with high conditioning and unimodal: Ellipsoidal, Discus, Bent Cigar, Sharp Ridge, and Different Powers.

- **F15-F19**: Multi-modal functions with adequate global structure: Rastrigin, Weierstrass, Schaffer's F7, ill-conditioned Schaffer's F7, and Composite Griewank-Rosenbrock.

- **F20-F24**: Multi-modal functions with weak global structure: Schwefel, Gallagher's Gaussian 101-me Peaks Function, Gallagher's Gaussian 21-hi Peaks Function, Katsuura, and Lunacek bi-Rastrigin.

For detailed definitions of the problems, we refer to Hansen et al. (2010). We provide the definition of Sphere (F1) for discussion in the paper.

$$f_{\text{Sphere}} : \mathcal{R}^n \to \mathcal{R}, \mathbf{x} \mapsto \| z \|^2 + f_{\text{opt}},$$

where $\mathbf{z} = \mathbf{x} - \mathbf{x}_{\text{opt}}$ and $\mathcal{R} \in [-5, 5]$.

## D    PROMPT DESIGN

Prompt design is critical to all LLM-driven optimization approaches. In this section, we first describe the general structure of the prompts used in all our experiments, and then we further discuss the specific parts that are customized for each of the three experiments (Section 4-6).

Our prompts implement the concept of the (1+1) elitist search strategy. All initial prompts consist of four components, each marked by ■: *Task Description*, including a Role instruction as well as the Problem and Task Descriptions, the Reference Code, and the instructions specifying the *Expected Outputs*. During the optimization loop, the prompt includes two additional components, each marked with □: *Parent Heuristic(s)*, including the linguistic description (together with its fitness value *Note*) and updated Reference code, as well as the *Strategy* sepcifying the instruction to the LLM for deriving a new solution based on the parent. The *italicized* text enclosed within /∗ and ∗/ provides auxiliary explanations, while **the regular text marked with ▷ is the actual prompt**. The general template is provided as follows:

---

**The Prompt Template**

**■ Role**
▷ You are an excellent Python programmer. You are a Python expert working on a new optimization algorithm.

**■ Problem and Task Descriptions**
/∗ *Provide here, the descriptions of the black-box optimization problems, along with the specific task assigned to the LLM.* ∗/

**□ Linguistic Descriptions of the Parent Heuristic**
/∗ *Include the descriptions (also the fitness value) of the generated parent heuristic here. This component does not appear in the initial prompt. It is partially produced by the LLM during the optimization loop and organized according to a template, and then used together with the reference code section that follows.*∗/

**■ Reference Code**
/∗ *Place the reference code here. The code should serve as a feasible solver for the black-box problem, meaning it is free of errors at a minimum.*∗/

**□ Strategy**
/∗ *This component defines the instructions given to the LLM for deriving a new solution from the provided parent heuristic.*∗/

**■ Expected Outputs**
▷ Provide the Python code and also give it a one-line description, describing the main idea.
Give the response in the format:
# Description: <short-description>
# Code:
```python

```

---

With the overall prompt template, we first introduce the template of the linguistic descriptions of the parent heuristic, a shared component across all experiments:

---

**The Template for Descriptions of the Parent**

☐ **Linguistic Descriptions of the Parent Heuristic**
▷ The current population of algorithms already evaluated (name, description, score) is:
/∗ *Names and Descriptions of the parent given by the LLM*∗/ (Score: /∗ *the fitness value*∗/)
The selected solution to update is:
/∗ *Descriptions of the parent given by the LLM*∗/
With code:
/∗ *Immediately followed by the component Reference Code.*∗/

---

Next, we introduce the two task-specific description components for the pbo and bbob suites. The task description for these two suites shares much common content, **with the main differences (highlighted in bold)** lying in the problem characteristics (pseudo-Boolean vs. continuous), aligning with the variable types (Boolean variables vs. bounded continuous space). **Note that the suite and problem names are omitted from the prompt to ensure proper alignment with the properties of BBO.** Details of the prompts are given as follow:

---

**Prompt Components for Pbo Benchmark Suite - Part 1**

■ **Problem and Task Descriptions**
▷ Your task is to develop a novel heuristic optimization algorithm for **pseudo-boolean maximization problems**. The optimization algorithm should work on different instances of noiseless unconstrained functions. Your task is to write the optimization algorithm in Python code. Each optimization problem has a search space **consisting of binary decision variables that can take values 0 or 1**. The dimensionality can be varied.
The code should contain an `__init__(self, budget, dim, seed)` function with optional additional arguments and the function ` def __call__(self, func)`, which should optimize the black box function `func` using `self.budget` function evaluations. The func() can only be called as many times as the budget allows, not more.

---

**Prompt Components for Bbob Suite - Part 1**

■ **Problem and Task Descriptions**
▷ Your task is to write a heuristic optimization algorithm for **continuous problems**. The optimization algorithm should work on different instances of noiseless unconstrained functions. Your task is to write an optimization algorithm in Python. Each of the optimization functions has a **search space between -5.0 (lower bound) and 5.0 (upper bound)**. The dimensionality can be varied.
The code should contain an `__init__(self, budget, dim, seed)` function with optional additional arguments and the function ` def __call__(self, func)`, which should optimize the black box function `func` using `self.budget` function evaluations. The func() can only be called as many times as the budget allows, not more.

---

D.1 Initial Prompts for Pbo and Bbob Suites

The initial reference code for all LLM-driven optimizers except for BAG is given as follow:

**Prompt Components for Pbo Benchmark Suite - Part 2**

■ **Reference Code for the Initial Prompt**

▷ An example of such code (a simple random search), is as follows:

```python
import numpy as np
class RandomSearch:
    def __init__(self, budget, dim, seed=None):
        self.budget = budget
        self.dim = dim
        self.rng = np.random.default_rng(seed)

    def __call__(self, func):
        f_opt = -np.Inf
        x_opt = None
        while func.state.evaluations < self.budget:
            x = self.rng.integers(0, 2, size=self.dim)
            f = func(x)
            if f > f_opt:
                f_opt = f
                x_opt = x
            if f_opt >= func.optimum.y:
                break
        return f_opt, x_opt
```

**Prompt Components for Bbob Suite - Part 2**

■ **Reference Code for Initial Prompt**

▷ An example of such code (a simple random search), is as follows:

```python
import numpy as np
class RandomSearch:
    def __init__(self, budget, dim, seed=None):
        self.budget = budget
        self.dim = dim
        self.rng = np.random.default_rng(seed)

    def __call__(self, func):
        f_opt = np.Inf
        x_opt = None
        for i in range(self.budget):
            x = self.rng.uniform(func.bounds.lb, func.bounds.ub)
            f = func(x)
            if f < f_opt:
                f_opt = f
                x_opt = x
            if f_opt <= func.optimum.y:
                break
        return f_opt, x_opt
```

By default, we adopt (global) random search as the code-formatting guideline for EoH, LLaMEA, and ReEvo, due to its simplicity.

## D.2   PROMPTS DESIGN FOR THE REFINEMENT-ONLY OPTIMIZATION

In this section, we present the prompts used in the experiments to guide the LLM toward specific search regions (see Section 5) and also in the final BAG optimizer (see section 6). In this setting, each initial prompt and each subsequent prompt that refers to the $\mathbf{A}_{\text{bench}}$ contains a promising starting code, and the LLM is required to refine this code as specified in the strategy block. An example of a reference code block and the corresponding strategy block for a pbo problem is shown below:

---

**Prompt Components for the Refinement Experiment**

■ **Reference Code** /∗ *Here, the greedy hill climber is used as the example.*∗/
▷ An example of such code for a good optimization algorithm is as follows:
```python
import numpy as np
class GreedyHillClimber:
    def __init__(self, budget, dim, seed=None):
        self.budget = budget
        self.dim = dim
        self.rng = np.random.default_rng(seed)

    def __call__(self, func):
        self.mutation_rate = (1.0 / self.dim)
        x = self.rng.integers(0, 2, self.dim)
        fx = func(x)
        idx = 0
        while func.state.evaluations < self.budget:
            y = x.copy()
            y[idx] = 1 - y[idx]   # flip bits for 0/1 domain
            idx = (idx + 1) % self.dim
            fy = func(y)
            if fy > fx:
                x, fx = y, fy
            if fy >= func.optimum.y:
                break
        return fx, x
```

□ **Strategy**
▷ Refine the example algorithm to improve its performance on the given task. Focus only on algorithmic changes, not formatting or comments.

---

## D.3   COMMON STRATEGIES

Below we list the two common strategies that are used in all three experiments for BAG (Sections 4, 5, 6). This is to say, besides the intial and refinement iterations, BAG dynamically chooses one of two strategies to be the strategy for the offspring (see lines 11 to 14 of Algorithm 1).

---

**Prompt Components - Strategy**

□ **Strategy 1**
▷ Refine the strategy of the selected solution to improve it.

□ **Strategy 2**
▷ Generate a new algorithm that is different from the algorithms you have tried before.

---

# E ADDITIONAL INFORMATION ON EXPERIMENTAL SETUP

**Details on the LLMs** We conduct the token-wise explanation experiments on two open-source LLMs and evaluate all LLM-driven optimization methods on three proprietary LLMs, as summarized in Table 3. All models are used with their default (sampling) parameters provided by the respective versions. The open-source models are loaded using the transformers library provided by Hugging Face (Wolf et al., 2020). They are selected because two of the proprietary models are built upon the similar underlying techniques.

Table 3: The used LLMs

| Open-source models used in Section 4. Models acquired from Hugging Face. | | |
| --- | --- | --- |
| **Name** | **Version** | **Provider** |
| Gemma 3 | google/gemma-3-27b-it | Google DeepMind |
| Qwen 2.5 Coder | Qwen/Qwen2.5-Coder-14B-Instruct | Alibaba Cloud |
| Proprietary models used in Sections 5 and 6 | | |
| **Name** | **Version** | **Provider** |
| Gemini 2.0 Flash | gemini-2.0-flash-001 | Google DeepMind |
| GPT 5 Nano | gpt-5-nano-2025-08-07 | OpenAI |
| Qwen3 Coder Flash | qwen3-coder-flash-2025-07-28 | Alibaba Cloud |

**Setups of pbo and bbob suites** For each problem (function) in pbo and bbob (see Appendix C), we evaluate every generated algorithm on five instances of the problem to compute the mean AUC performance. To ensure fairness in CPU time, we set an evaluation timeout for each instance. Table 4 summarizes the setup of the pbo and bbob problem suites.

Table 4: Setups of the pbo and bbob suites

| Suite | Dimensionality | Budget | Evaluation timeout per instance |
| --- | --- | --- | --- |
| pbo | 100 | 1,000,000 | 600 seconds |
| bbob | 5 | 10,000 | 600 seconds |

**Setups of software** We use the default implementations and hyperparameters for EoH[1] (see Section 4.1 of Liu et al. (2024a)), LLaMEA[2] (see Section IV of van Stein & Bäck (2024)), and ReEvo[3] (see Appendix C of Ye et al. (2024)). The internal prompt engineering of all three baseline models is kept unchanged. For fairness and consistency, we provide only the *Task Description* along with the initial random search code to each method. We use the default implementation of AttnLRP[4] for the experiment on token-wise analysis of prompt design.

**Hardware Specifications** The token-wise analysis experiment is conducted on a server equipped with two NVIDIA GeForce RTX 3090 GPUs and an Intel Xeon Silver 4214R CPU. For large-scale benchmarking on the pbo and bbob problem suites, each problem (function) is assigned a single core of an AMD EPYC 7662 CPU for every LLM-driven optimizer. Importantly, as discussed in Section 6, the final experimental results are independent of the underlying hardware.

---

[1]`https://github.com/FeiLiu36/EoH`
[2]`https://github.com/XAI-liacs/LLaMEA`
[3]`https://github.com/ai4co/reevo`
[4]`https://github.com/rachtibat/LRP-eXplains-Transformers`

# F  IMPACT OF THE FREQUENCY FACTOR

In this section, we study the impact of the frequency factor $q$ in Algorithm 1. Specifically, we compare the AUC values obtained with different $q \in \{1, 5, 10, 20, 40, 100\}$. Figures 7 and 8 present the averaged (normalized) performance of the three LLMs on the pbo and bbob benchmarks, respectively. These results show that BAG achieves the best and most robust results when $q = 10$. Under this setting, BAG is expected to utilize each algorithm in $\mathbf{A}_{bench}$ (of size 5) approximately twice on average.

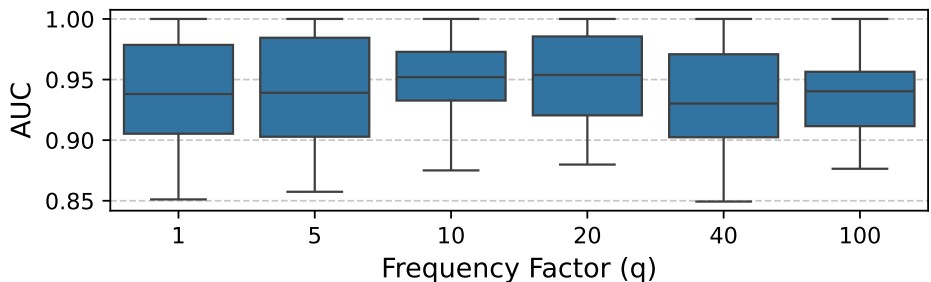

Figure 7: Average performance of BAG with different frequency factor $q$. The results are aggregated across three LLMs for pbo benchmarks.

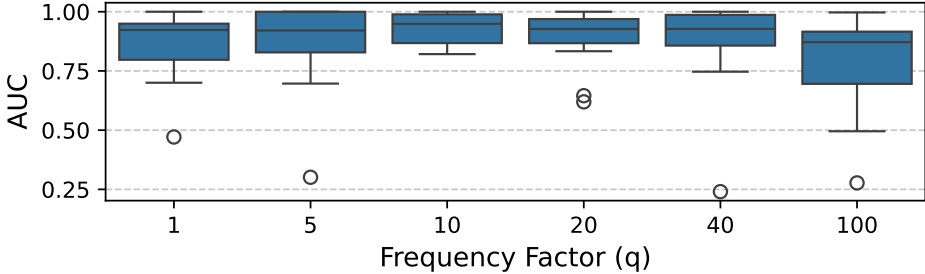

Figure 8: Average performance of BAG with different frequency factor $q$. The results are aggregated across three LLMs for bbob benchmarks.

## G  ADDITIONAL REFINEMENT RESULTS

We provide additional results comparing the AUC values of the algorithms obtained by EoH, LLaMEA, ReEvo, and our refinement strategy, which explicitly queries LLMs to refine the five provided benchmark codes, respectively. In addition to the results introduced in Section 5, we report further experiments on LeadingOnes (F2), one variant for each (F10, F17), an Ising model (F19), and MIVS (F22), covering both theory-oriented and practical scenarios.

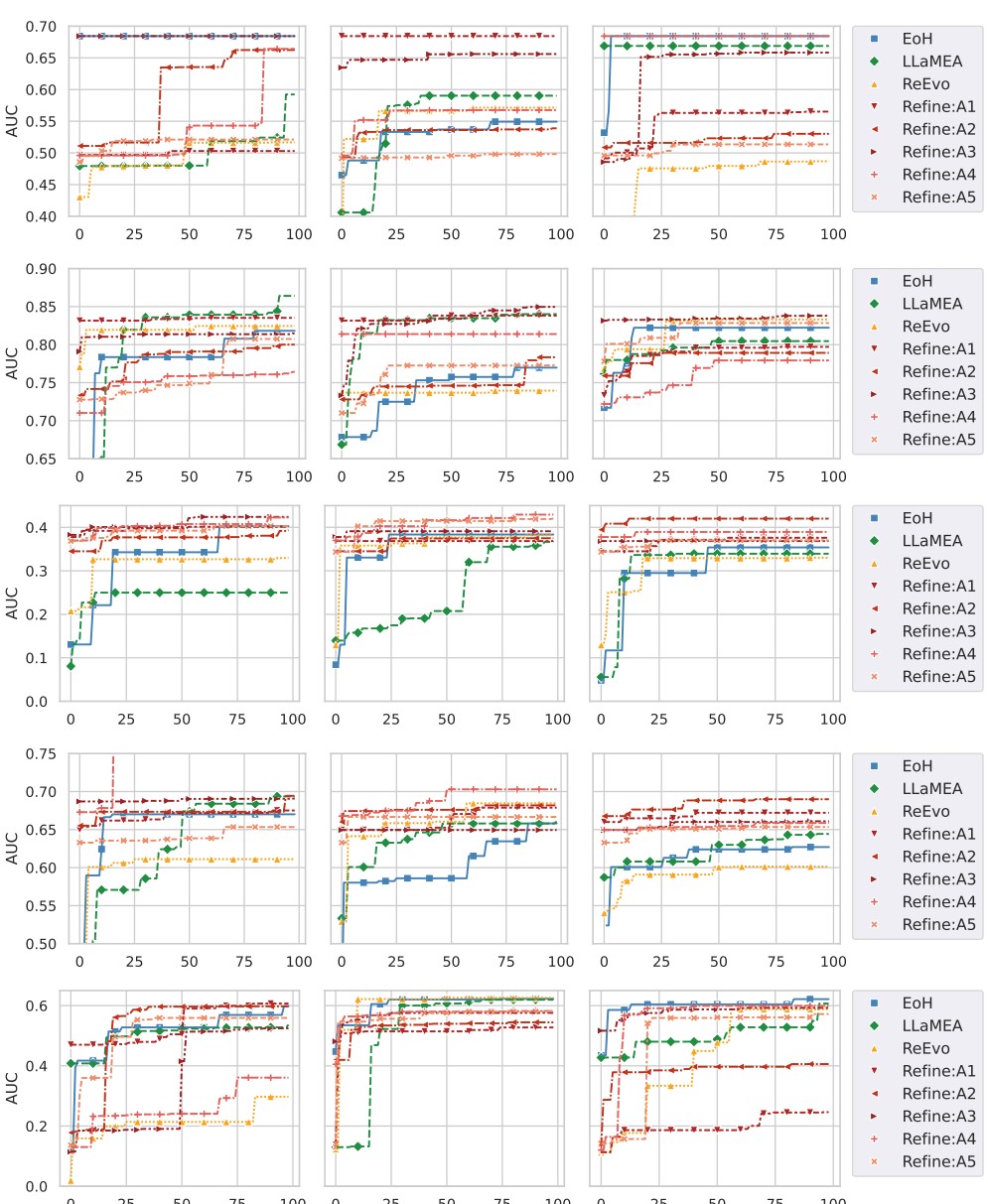

Figure 9: AUC values of the algorithms obtained by LLM-driven approaches on F2, F10, F17, F19, and F22 of the pbo suite (from Top to Bottom). The $x$-axis represents the cumulative number of algorithms generated by the LLM, and the $y$-axis indicates the best-so-far AUC value. The results are obtained using GPT 5 Nano, Gemini 2.0 Flash, and Qwen3 Coder Flash (from Left to Right).

# H    ANALYSIS OF IMPACT OF BENCHMARK(-INDUCED) ALGORITHM TO GENERATED HEURISTICS

To analyze the impact of benchmark algorithm on the generated heuristics, we use CodeBLEU Ren et al. (2020), a metric for measuring the similarity (relevance) between two pieces of code. Code-BLEU was originally proposed to evaluate the logical and structural quality of generated code by comparing it with a reference implementation. Given a reference code snippet (R) and a generated code snippet (S), CodeBLEU computes a weighted sum of four sub-metrics derived from R and S. These include: 1) BLEU, which measures exact sequential token matches between R and S; 2) weighted n-gram matching of BLEU, which emphasizes the influence of key tokens such as data types and keywords; 3) abstract syntax tree (AST) matching, which captures structural consistency across code lines; and 4) data-flow matching, which evaluates semantic similarity based on variable dependencies, such as whether later code correctly references earlier definitions of variable.

In our setting, CodeBLEU[5] can provide a quantitative measure of how much textual, structural, and semantic information the generated heuristics inherit from earlier heuristics. This allows us to assess how the injected benchmark algorithm influences solutions produced at later stages of the evolutionary search process.

In our setup, new algorithms are generated in a (1+1) evolutionary manner, resulting in an ordered sequence with one-directional influence: later algorithms inherit information from their promising predecessors, but not the other way around. To check this property, for each problem and each LLM, we compute pairwise similarity scores for all ordered pairs of algorithms within the same search trajectory of BAG using an upper-triangular scheme. Specifically, given a sequence of algorithms $[\mathcal{A}_1, \ldots, \mathcal{A}_N]$, we iterate over indices $i \in [1, N-1]$ and, for each $i$, compute the similarity between $\mathcal{A}_i$ and all subsequent heuristics $\mathcal{A}_j$ where $j \in [i+1, N]$. This procedure evaluates each valid pair exactly once while avoiding redundant and self-comparisons, and preserves the temporal direction of influence inherent in the generation process of BAG.

---

[5]We use the open-source implementation:https://github.com/k4black/codebleu.

# I ADDITION COMPARISON RESULTS

We present the detailed performance of our BAG method on each individual benchmark problem, compared to EoH, LLaMEA, and ReEvo. The results include the normalized AUC values, the corresponding ranks based on AUC, and the convergence process of the obtained AUC throughout the search process of each method.

Table 5: The best normalized (higher is better) AUC achieved by four LLM-driven approaches on 23 pbo problems. Results are obtained using Gemini 2.0 Flash. Each normalized AUC value is followed by its corresponding rank in brackets. The best entries are underlined.

| Function ID | BAG | EoH | LLaMEA | ReEvo |
|:---:|:---:|:---:|:---:|:---:|
| F1 | 0.9749 (2) | 0.9695 (4) | 1.0000 (1) | 0.9723 (3) |
| F2 | 0.8056 (3) | 1.0000 (1) | 0.8654 (2) | 0.7549 (4) |
| F3 | 1.0000 (1) | 0.9871 (3) | 0.9882 (2) | 0.9643 (4) |
| F4 | 0.9863 (3) | 1.0000 (1) | 0.9853 (4) | 0.9974 (2) |
| F5 | 0.9716 (4) | 1.0000 (1) | 0.9840 (2) | 0.9840 (2) |
| F6 | 1.0000 (1) | 0.9996 (2) | 0.9957 (3) | 0.9912 (4) |
| F7 | 1.0000 (1) | 0.9152 (4) | 0.9499 (3) | 0.9862 (2) |
| F8 | 0.9654 (4) | 1.0000 (1) | 0.9765 (2) | 0.9710 (3) |
| F9 | 1.0000 (1) | 0.9824 (4) | 0.9888 (2) | 0.9842 (3) |
| F10 | 0.9727 (2) | 0.9469 (4) | 1.0000 (1) | 0.9542 (3) |
| F11 | 0.9747 (3) | 1.0000 (1) | 0.9898 (2) | 0.9718 (4) |
| F12 | 0.7565 (3) | 0.9727 (2) | 0.7467 (4) | 1.0000 (1) |
| F13 | 0.8682 (2) | 0.8350 (4) | 0.8449 (3) | 1.0000 (1) |
| F14 | 0.8761 (2) | 0.7662 (4) | 1.0000 (1) | 0.8359 (3) |
| F15 | 1.0000 (1) | 0.8776 (3) | 0.9409 (2) | 0.8676 (4) |
| F16 | 1.0000 (1) | 0.9370 (4) | 0.9810 (3) | 0.9913 (2) |
| F17 | 1.0000 (1) | 0.9860 (2) | 0.6113 (4) | 0.8061 (3) |
| F18 | 0.9586 (2) | 0.9572 (3) | 0.9492 (4) | 1.0000 (1) |
| F19 | 0.9550 (3) | 0.9660 (2) | 1.0000 (1) | 0.8809 (4) |
| F20 | 0.9405 (3) | 0.9165 (4) | 0.9475 (2) | 1.0000 (1) |
| F21 | 0.9908 (2) | 0.9908 (2) | 1.0000 (1) | 0.8560 (4) |
| F22 | 1.0000 (1) | 0.9778 (2) | 0.8771 (3) | 0.4862 (4) |
| F23 | 1.0000 (1) | 0.9667 (4) | 0.9932 (2) | 0.9932 (2) |
| Mean | 0.9564 | 0.9543 | 0.9398 | 0.9239 |
| Std | 0.0652 | 0.0578 | 0.0938 | 0.1169 |
| Average Rank | 2.00 | 2.67 | 2.38 | 2.83 |

Table 6: The best normalized (higher is better) AUC achieved by four LLM-driven approaches on 23 pbo problems. Results are obtained using GPT 5 Nano. Each normalized AUC value is followed by its corresponding rank in brackets. The best entries are underlined.

| Function ID | BAG | EoH | LLaMEA | ReEvo |
|:---:|:---:|:---:|:---:|:---:|
| F1 | 0.9883 (4) | 0.9916 (2) | 1.0000 (1) | 0.9915 (3) |
| F2 | 1.0000 (1) | 0.7840 (4) | 0.8425 (2) | 0.8156 (3) |
| F3 | 0.9949 (2) | 0.9707 (4) | 1.0000 (1) | 0.9797 (3) |
| F4 | 0.9374 (3) | 0.9217 (4) | 1.0000 (1) | 0.9562 (2) |
| F5 | 0.9485 (3) | 1.0000 (1) | 0.9196 (4) | 0.9833 (2) |
| F6 | 0.9770 (2) | 1.0000 (1) | 0.9744 (3) | 0.9580 (4) |
| F7 | 0.9225 (4) | 0.9877 (2) | 1.0000 (1) | 0.9292 (3) |
| F8 | 0.9929 (2) | 0.9528 (3) | 1.0000 (1) | 0.9439 (4) |
| F9 | 0.9916 (2) | 0.9806 (4) | 0.9858 (3) | 1.0000 (1) |
| F10 | 0.9274 (2) | 0.9163 (3) | 1.0000 (1) | 0.8803 (4) |
| F11 | 0.9341 (2) | 1.0000 (1) | 0.8326 (4) | 0.9339 (3) |
| F12 | 0.9966 (2) | 1.0000 (1) | 0.7270 (4) | 0.7679 (3) |
| F13 | 1.0000 (1) | 0.9967 (2) | 0.9914 (3) | 0.9356 (4) |
| F14 | 0.6954 (3) | 1.0000 (1) | 0.8333 (2) | 0.5391 (4) |
| F15 | 1.0000 (1) | 0.9076 (2) | 0.7601 (4) | 0.8406 (3) |
| F16 | 1.0000 (1) | 0.8737 (4) | 0.9549 (3) | 0.9623 (2) |
| F17 | 1.0000 (1) | 0.9736 (2) | 0.9309 (4) | 0.9697 (3) |
| F18 | 0.9160 (4) | 0.9569 (3) | 1.0000 (1) | 0.9721 (2) |
| F19 | 0.9489 (4) | 0.9614 (3) | 0.9652 (2) | 1.0000 (1) |
| F20 | 0.9669 (3) | 1.0000 (1) | 0.9571 (4) | 0.9769 (2) |
| F21 | 0.9687 (3) | 1.0000 (1) | 0.9373 (4) | 0.9724 (2) |
| F22 | 0.9230 (4) | 0.9945 (2) | 0.9893 (3) | 1.0000 (1) |
| F23 | 0.9006 (2) | 0.8851 (3) | 0.7828 (4) | 1.0000 (1) |
| Mean | 0.9535 | 0.9589 | 0.9298 | 0.9264 |
| Std | 0.0638 | 0.0537 | 0.0854 | 0.1024 |
| Average Rank | 2.42 | 2.29 | 2.62 | 2.67 |

Table 7: The best normalized (higher is better) AUC achieved by four LLM-driven approaches on 23 pbo problems. Results are obtained using Qwen3 Coder Flash. Each normalized AUC value is followed by its corresponding rank in brackets. The best entries are underlined.

| Function ID | BAG | EoH | LLaMEA | ReEvo |
|---|---|---|---|---|
| F1 | 0.9734 (2) | 1.0000 (1) | 0.9699 (3) | 0.9442 (4) |
| F2 | 0.9313 (3) | 1.0000 (1) | 0.9773 (2) | 0.7115 (4) |
| F3 | 0.9936 (3) | 0.9999 (2) | 1.0000 (1) | 0.9628 (4) |
| F4 | 0.9938 (2) | 1.0000 (1) | 0.9444 (4) | 0.9513 (3) |
| F5 | 1.0000 (1) | 0.9813 (2) | 0.9526 (3) | 0.9321 (4) |
| F6 | 0.9817 (3) | 0.9865 (2) | 0.9706 (4) | 1.0000 (1) |
| F7 | 1.0000 (1) | 0.9578 (3) | 0.9778 (2) | 0.9306 (4) |
| F8 | 1.0000 (1) | 0.9546 (3) | 0.9161 (4) | 0.9694 (2) |
| F9 | 1.0000 (1) | 0.9798 (3) | 0.9621 (4) | 0.9967 (2) |
| F10 | 0.9912 (2) | 0.9870 (3) | 0.9659 (4) | 1.0000 (1) |
| F11 | 0.9513 (2) | 1.0000 (1) | 0.9311 (3) | 0.8351 (4) |
| F12 | 0.9460 (3) | 1.0000 (1) | 0.9601 (2) | 0.8285 (4) |
| F13 | 1.0000 (1) | 0.9982 (2) | 0.9842 (3) | 0.9682 (4) |
| F14 | 0.8515 (3) | 1.0000 (1) | 0.9600 (2) | 0.8152 (4) |
| F15 | 1.0000 (1) | 0.9038 (4) | 0.9040 (3) | 0.9087 (2) |
| F16 | 1.0000 (1) | 0.8654 (2) | 0.7675 (4) | 0.8558 (3) |
| F17 | 1.0000 (1) | 0.9291 (2) | 0.8910 (3) | 0.8669 (4) |
| F18 | 0.9953 (2) | 0.9851 (3) | 1.0000 (1) | 0.8488 (4) |
| F19 | 1.0000 (1) | 0.9220 (3) | 0.9474 (2) | 0.8842 (4) |
| F20 | 0.8847 (2) | 0.8711 (3) | 1.0000 (1) | 0.8183 (4) |
| F21 | 1.0000 (1) | 0.9837 (2) | 0.8231 (4) | 0.8681 (3) |
| F22 | 0.9448 (4) | 1.0000 (1) | 0.9763 (2) | 0.9461 (3) |
| F23 | 0.9891 (3) | 0.9966 (2) | 1.0000 (1) | 0.9428 (4) |
| Mean | 0.9751 | 0.9697 | 0.947 | 0.9037 |
| Std | 0.0393 | 0.0414 | 0.0556 | 0.0718 |
| Average Rank | 1.88 | 2.08 | 2.71 | 3.33 |

Table 8: The best normalized (higher is better) AUC achieved by four LLM-driven approaches on 24 bbob problems. Results are obtained using Gemini 2.0 Flash. Each normalized AUC value is followed by its corresponding rank in brackets. The best entries are underlined.

| Function ID | BAG | EoH | LLaMEA | ReEvo |
|---|---|---|---|---|
| F1 | 1.0000 (1) | 0.9901 (2) | 0.9876 (3) | 0.9850 (4) |
| F2 | 0.9711 (3) | 1.0000 (1) | 0.9713 (2) | 0.9148 (4) |
| F3 | 0.8787 (3) | 1.0000 (1) | 0.9984 (2) | 0.8437 (4) |
| F4 | 0.8338 (2) | 0.6228 (3) | 1.0000 (1) | 0.4922 (4) |
| F5 | 0.9983 (3) | 1.0000 (1) | 0.9993 (2) | 0.9896 (4) |
| F6 | 0.9847 (2) | 1.0000 (1) | 0.9721 (3) | 0.9627 (4) |
| F7 | 1.0000 (1) | 0.9466 (3) | 0.9158 (4) | 0.9718 (2) |
| F8 | 1.0000 (1) | 0.8066 (4) | 0.9664 (3) | 0.9824 (2) |
| F9 | 1.0000 (1) | 0.6241 (3) | 0.5308 (4) | 0.7263 (2) |
| F10 | 0.9577 (2) | 1.0000 (1) | 0.5922 (4) | 0.8079 (3) |
| F11 | 0.9238 (4) | 0.9607 (2) | 1.0000 (1) | 0.9405 (3) |
| F12 | 0.9794 (2) | 1.0000 (1) | 0.6469 (4) | 0.8572 (3) |
| F13 | 0.8878 (4) | 0.9870 (2) | 1.0000 (1) | 0.9715 (3) |
| F14 | 0.9899 (2) | 0.9503 (3) | 1.0000 (1) | 0.9430 (4) |
| F15 | 1.0000 (1) | 0.6262 (3) | 0.5939 (4) | 0.7565 (2) |
| F16 | 0.7911 (2) | 0.7089 (3) | 1.0000 (1) | 0.2889 (4) |
| F17 | 0.9789 (3) | 0.9908 (2) | 1.0000 (1) | 0.9743 (4) |
| F18 | 1.0000 (1) | 0.7248 (3) | 0.8356 (2) | 0.6963 (4) |
| F19 | 0.6239 (2) | 0.5441 (4) | 0.6178 (3) | 1.0000 (1) |
| F20 | 0.7606 (4) | 1.0000 (1) | 0.7735 (3) | 0.8670 (2) |
| F21 | 0.9976 (2) | 0.9115 (4) | 0.9120 (3) | 1.0000 (1) |
| F22 | 1.0000 (1) | 0.9596 (2) | 0.7053 (4) | 0.8429 (3) |
| F23 | 0.9174 (2) | 0.7581 (4) | 1.0000 (1) | 0.8092 (3) |
| F24 | 0.9542 (3) | 0.9664 (2) | 1.0000 (1) | 0.9421 (4) |
| Mean | 0.9345 | 0.8783 | 0.8758 | 0.8569 |
| Std | 0.0942 | 0.1513 | 0.1626 | 0.1684 |
| Average Rank | 2.12 | 2.32 | 2.44 | 3.12 |

Table 9: The best normalized (higher is better) AUC achieved by four LLM-driven approaches on 24 bbob problems. Results are obtained using GPT 5 Nano. Each normalized AUC value is followed by its corresponding rank in brackets. The best entries are underlined.

| Function ID | BAG | EoH | LLaMEA | ReEvo |
|---|---|---|---|---|
| F1 | 0.9852 (3) | 1.0000 (1) | 0.9865 (2) | 0.9769 (4) |
| F2 | 1.0000 (1) | 0.9546 (3) | 0.9438 (4) | 0.9881 (2) |
| F3 | 1.0000 (1) | 0.6655 (2) | 0.5437 (3) | 0.4340 (4) |
| F4 | 0.8616 (2) | 0.7040 (3) | 1.0000 (1) | 0.1821 (4) |
| F5 | 0.9992 (3) | 0.9996 (2) | 0.9991 (4) | 1.0000 (1) |
| F6 | 0.9872 (4) | 0.9961 (3) | 0.9993 (2) | 1.0000 (1) |
| F7 | 1.0000 (1) | 0.5637 (3) | 0.9674 (2) | 0.4178 (4) |
| F8 | 0.8394 (2) | 0.2763 (3) | 1.0000 (1) | 0.2725 (4) |
| F9 | 0.8264 (2) | 0.3063 (4) | 1.0000 (1) | 0.5811 (3) |
| F10 | 1.0000 (1) | 0.0801 (4) | 0.7227 (2) | 0.0900 (3) |
| F11 | 1.0000 (1) | 0.1445 (3) | 0.2663 (2) | 0.1205 (4) |
| F12 | 1.0000 (1) | 0.2952 (4) | 0.3702 (3) | 0.3803 (2) |
| F13 | 1.0000 (1) | 0.3303 (3) | 0.3142 (4) | 0.3504 (2) |
| F14 | 1.0000 (1) | 0.7204 (3) | 0.8224 (2) | 0.6544 (4) |
| F15 | 0.8978 (2) | 0.4564 (4) | 1.0000 (1) | 0.4644 (3) |
| F16 | 1.0000 (1) | 0.8418 (2) | 0.5816 (4) | 0.7424 (3) |
| F17 | 0.9587 (2) | 0.6098 (4) | 1.0000 (1) | 0.6881 (3) |
| F18 | 1.0000 (1) | 0.5754 (3) | 0.4382 (4) | 0.6908 (2) |
| F19 | 0.9696 (2) | 0.9359 (3) | 1.0000 (1) | 0.9354 (4) |
| F20 | 1.0000 (1) | 0.6441 (3) | 0.6687 (2) | 0.4884 (4) |
| F21 | 0.9183 (3) | 0.8533 (4) | 1.0000 (1) | 0.9243 (2) |
| F22 | 1.0000 (1) | 0.8183 (3) | 0.9580 (2) | 0.6923 (4) |
| F23 | 1.0000 (1) | 0.7482 (2) | 0.6705 (3) | 0.6596 (4) |
| F24 | 0.1646 (2) | 1.0000 (1) | 0.1395 (3) | 0.1276 (4) |
| Mean | 0.9337 | 0.6467 | 0.7663 | 0.5776 |
| Std. | 0.1692 | 0.2816 | 0.2789 | 0.2935 |
| Average Rank | 1.64 | 2.92 | 2.28 | 3.16 |

Table 10: The best normalized (higher is better) AUC achieved by four LLM-driven approaches on 24 bbob problems. Results are obtained using Qwen3 Coder Flash. Each normalized AUC value is followed by its corresponding rank in brackets. The best entries are underlined.

| Function ID | BAG | EoH | LLaMEA | ReEvo |
|---|---|---|---|---|
| F1 | 0.9863 (3) | 0.9940 (2) | 1.0000 (1) | 0.9679 (4) |
| F2 | 1.0000 (1) | 0.9363 (4) | 0.9651 (3) | 0.9921 (2) |
| F3 | 1.0000 (1) | 0.7693 (3) | 0.9702 (2) | 0.1093 (4) |
| F4 | 0.9461 (2) | 0.5665 (4) | 0.8828 (3) | 1.0000 (1) |
| F5 | 0.9993 (2) | 0.9987 (3) | 0.9986 (4) | 1.0000 (1) |
| F6 | 1.0000 (1) | 0.8200 (2) | 0.7457 (3) | 0.6979 (4) |
| F7 | 0.9927 (2) | 0.8820 (3) | 1.0000 (1) | 0.6366 (4) |
| F8 | 1.0000 (1) | 0.7383 (3) | 0.9892 (2) | 0.5447 (4) |
| F9 | 0.9865 (2) | 0.8673 (3) | 0.5720 (4) | 1.0000 (1) |
| F10 | 0.9851 (2) | 0.3773 (4) | 0.7685 (3) | 1.0000 (1) |
| F11 | 1.0000 (1) | 0.3501 (4) | 0.3758 (3) | 0.5400 (2) |
| F12 | 1.0000 (1) | 0.7494 (3) | 0.4088 (4) | 0.9213 (2) |
| F13 | 1.0000 (1) | 0.4721 (3) | 0.6754 (2) | 0.3059 (4) |
| F14 | 1.0000 (1) | 0.8669 (2) | 0.8608 (3) | 0.6515 (4) |
| F15 | 1.0000 (1) | 0.5607 (3) | 0.7831 (2) | 0.4579 (4) |
| F16 | 0.7467 (3) | 0.8708 (2) | 0.7455 (4) | 1.0000 (1) |
| F17 | 0.8189 (3) | 0.8544 (2) | 0.6451 (4) | 1.0000 (1) |
| F18 | 1.0000 (1) | 0.5395 (2) | 0.4219 (4) | 0.4754 (3) |
| F19 | 1.0000 (1) | 0.9997 (2) | 0.9341 (3) | 0.8481 (4) |
| F20 | 0.7849 (2) | 1.0000 (1) | 0.6332 (4) | 0.7120 (3) |
| F21 | 0.9901 (2) | 1.0000 (1) | 0.8732 (3) | 0.4632 (4) |
| F22 | 0.9912 (2) | 1.0000 (1) | 0.8957 (3) | 0.5569 (4) |
| F23 | 0.8558 (3) | 0.8694 (2) | 0.7170 (4) | 1.0000 (1) |
| F24 | 1.0000 (1) | 0.8638 (3) | 0.8838 (2) | 0.8100 (4) |
| Mean | 0.9618 | 0.7894 | 0.7811 | 0.7371 |
| Std | 0.0744 | 0.2002 | 0.1895 | 0.2568 |
| Average Rank | 1.64 | 2.56 | 2.96 | 2.84 |

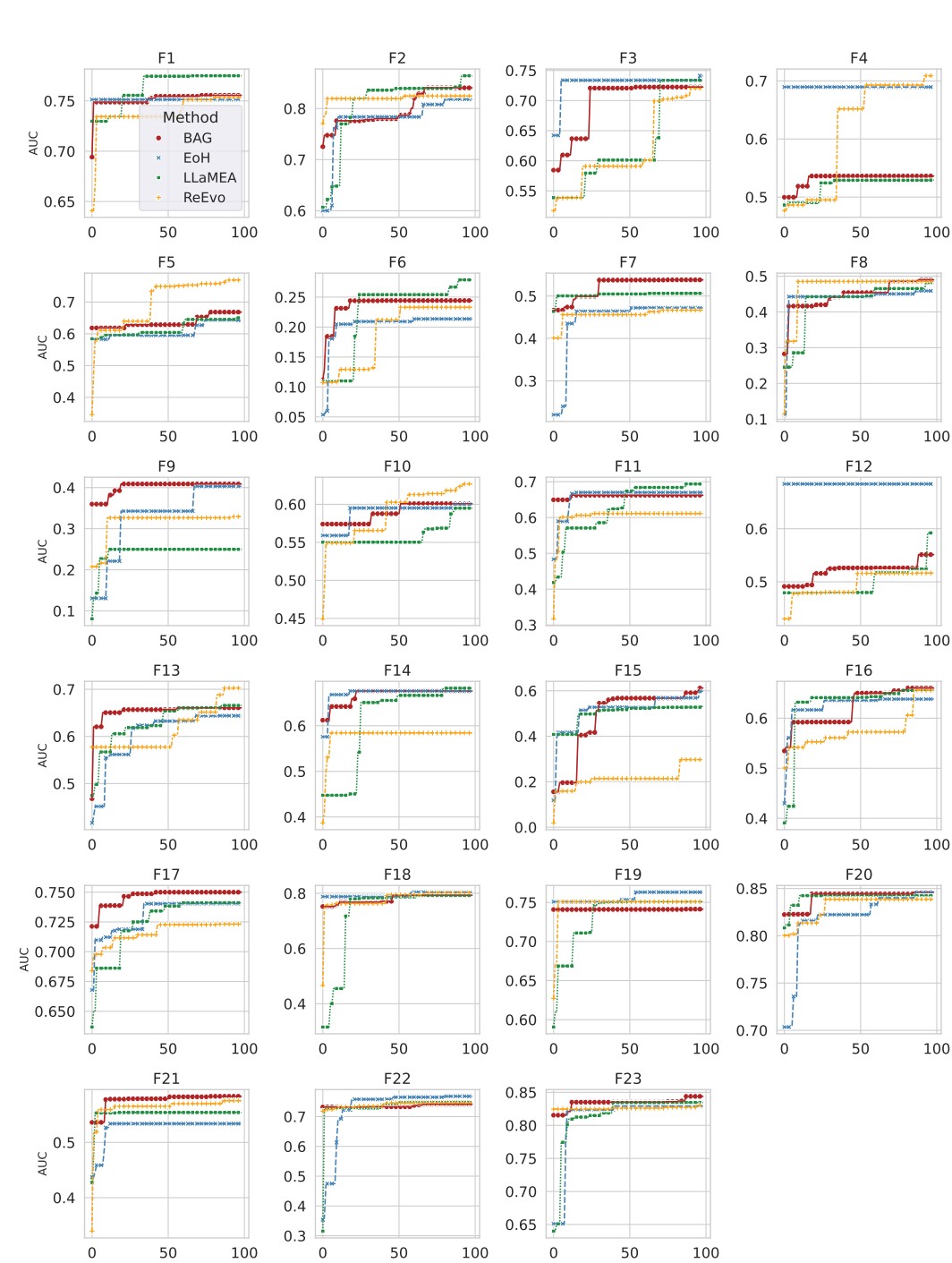

Figure 10: AUC values of the algorithms obtained by LLM-driven approaches on all pbo problems. The $x$-axis represents the cumulative number of algorithms generated by the LLM, and the $y$-axis indicates the best-so-far AUC value. The results are obtained using Gemini 2.0 Flash.

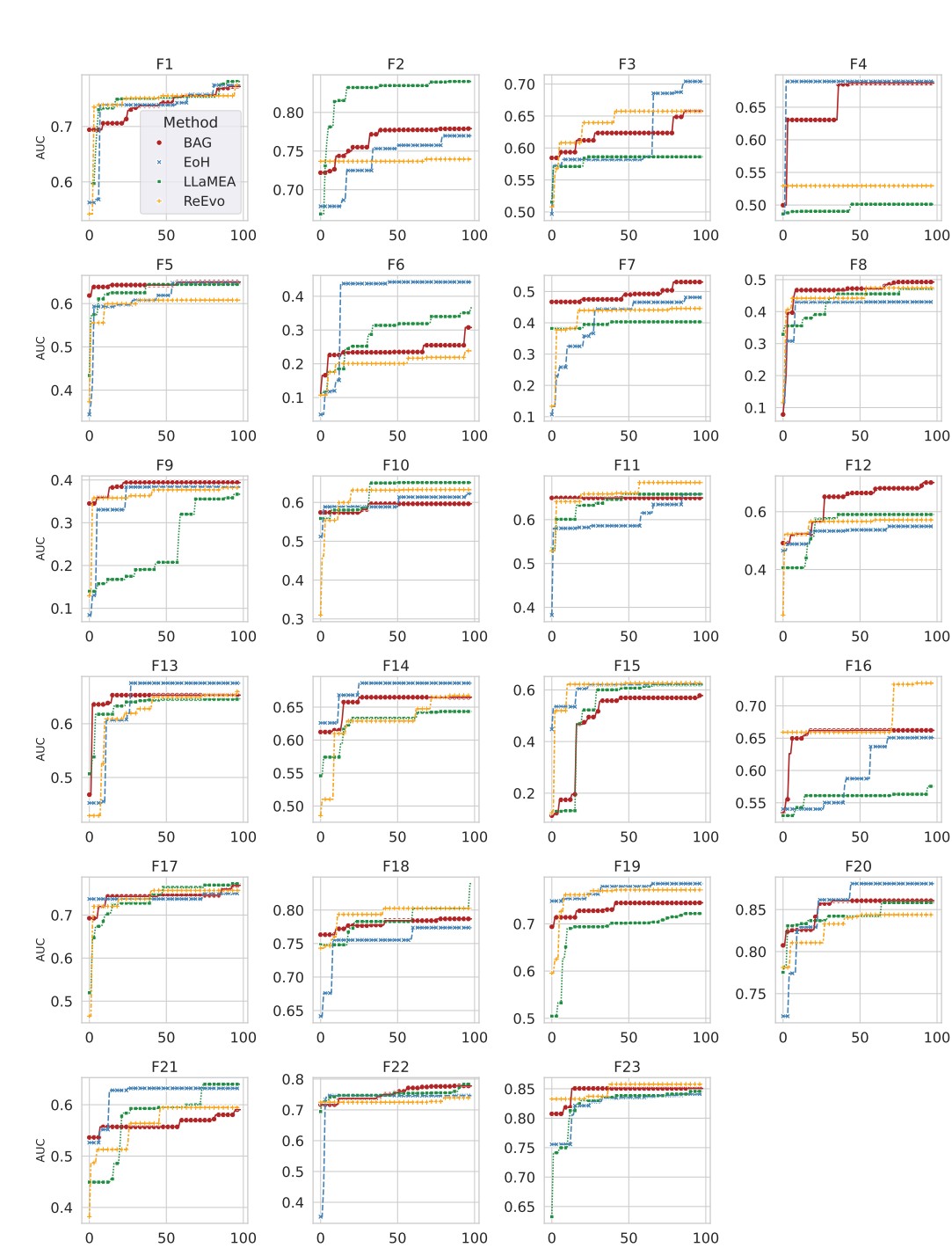

Figure 11: AUC values of the algorithms obtained by LLM-driven approaches on all pbo problems. The $x$-axis represents the cumulative number of algorithms generated by the LLM, and the $y$-axis indicates the best-so-far AUC value. The results are obtained using GPT 5 Nano.

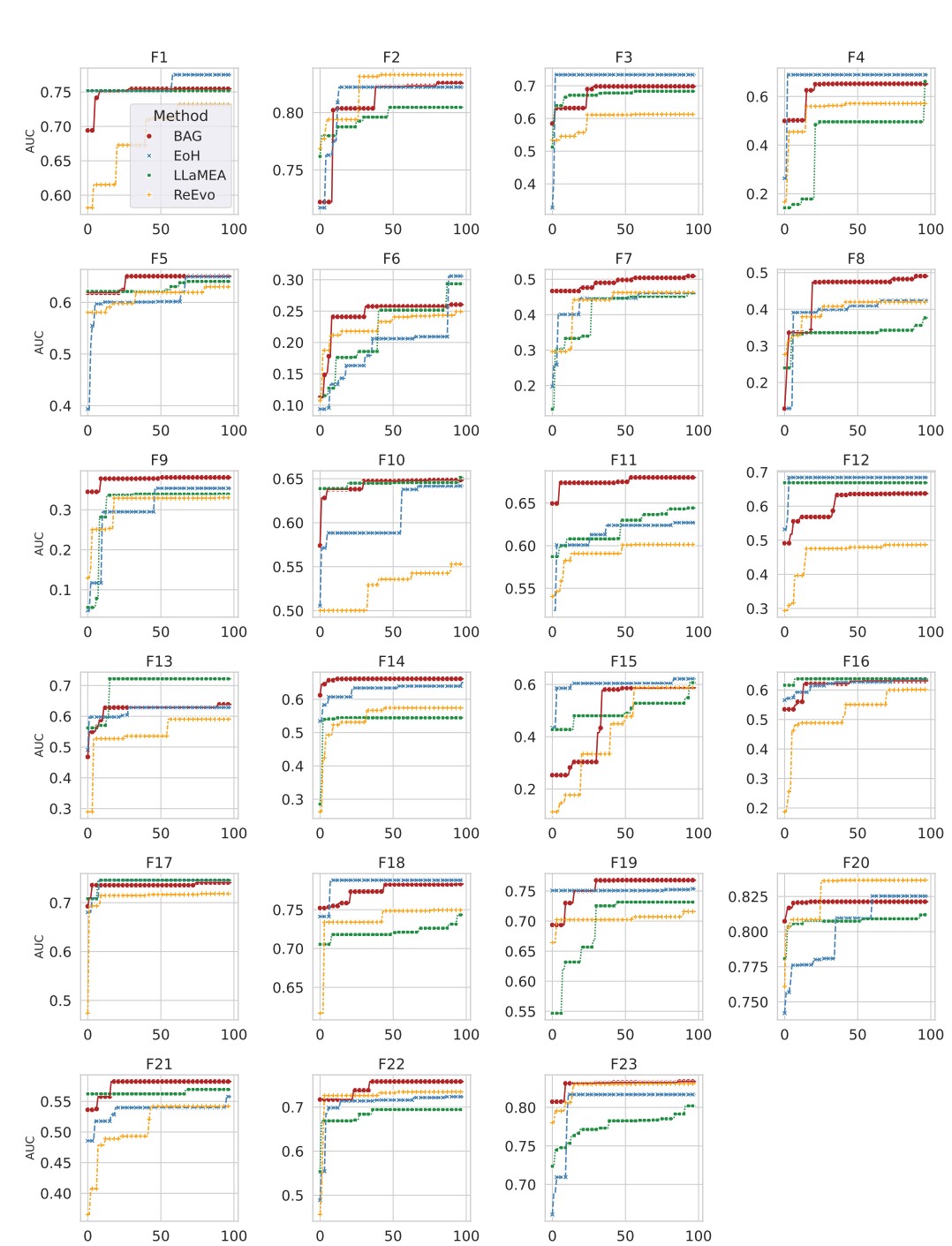

Figure 12: AUC values of the algorithms obtained by LLM-driven approaches on all pbo problems. The $x$-axis represents the cumulative number of algorithms generated by the LLM, and the $y$-axis indicates the best-so-far AUC value. The results are obtained using Qwen3 Coder Flash.

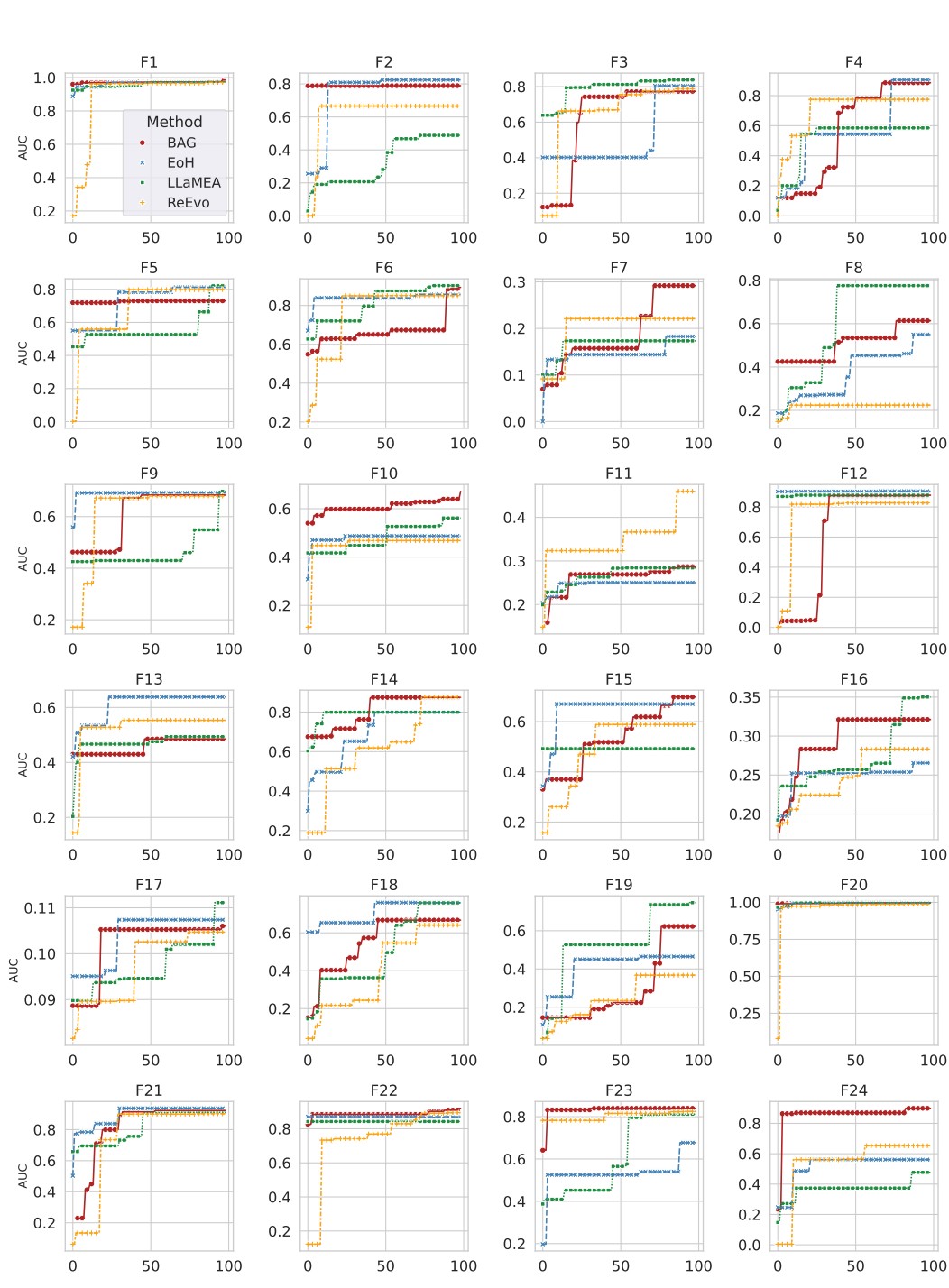

Figure 13: AUC values of the algorithms obtained by LLM-driven approaches on all bbob problems. The $x$-axis represents the cumulative number of algorithms generated by the LLM, and the $y$-axis indicates the best-so-far AUC value. The results are obtained using Gemini 2.0 Flash.

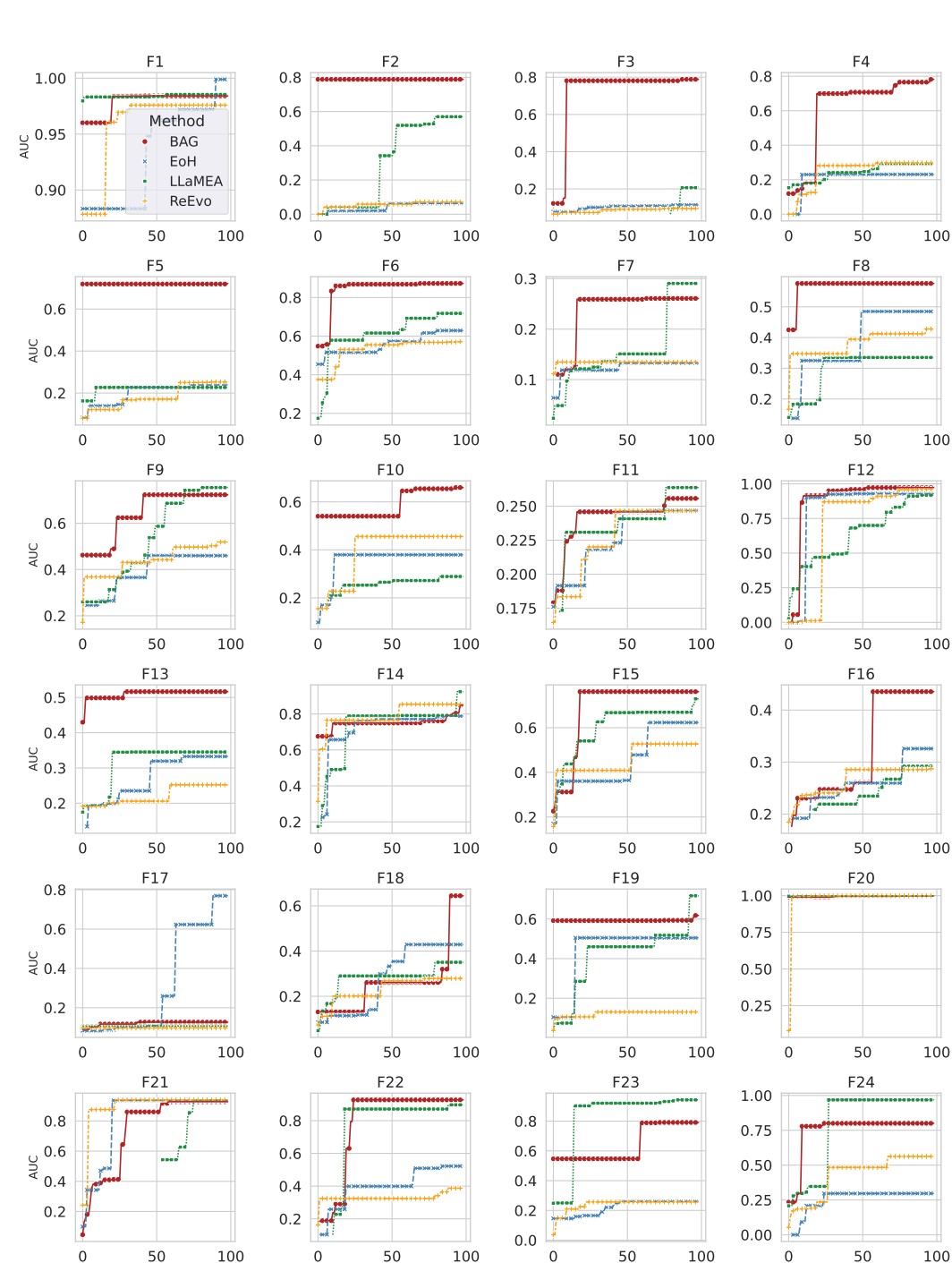

Figure 14: AUC values of the algorithms obtained by LLM-driven approaches on all bbob problems. The $x$-axis represents the cumulative number of algorithms generated by the LLM, and the $y$-axis indicates the best-so-far AUC value. The results are obtained using GPT 5 Nano.

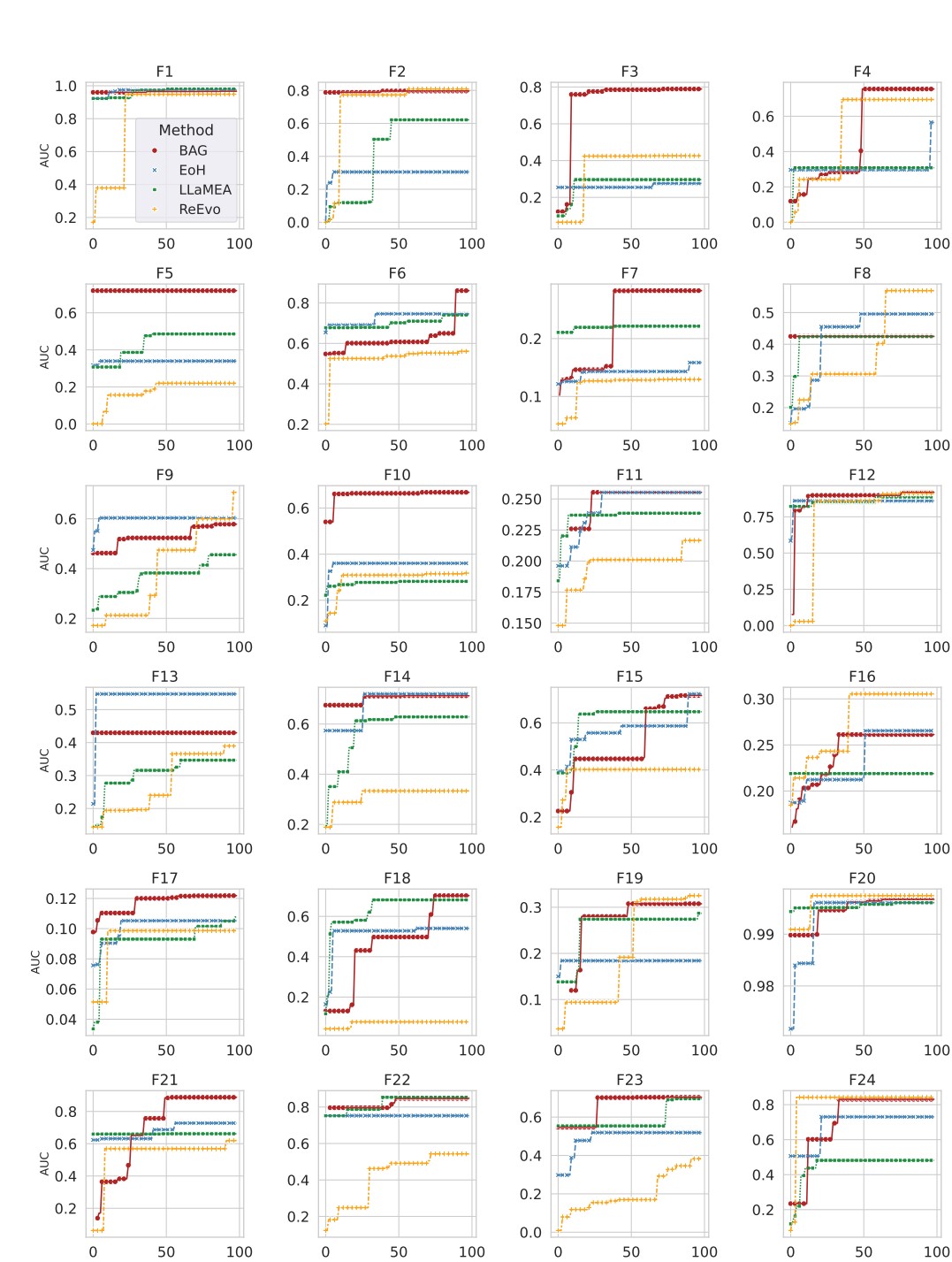

Figure 15: AUC values of the algorithms obtained by LLM-driven approaches on all bbob problems. The $x$-axis represents the cumulative number of algorithms generated by the LLM, and the $y$-axis indicates the best-so-far AUC value. The results are obtained using Qwen3 Coder Flash.

## J  RESULTS FOR TESTING GENERALIZATION

In this section, we evaluate the performance of the final algorithms obtained by LLM-driven optimization methods. Recall that during the searching process, the candidate algorithms are assessed on five problem instances. Here, we validate the algorithms' performance on a different set of five instances that are unseen during the search. The corresponding results are listed in Tables 11 and 12, aggregated across the three LLM models. The relative performance of the compared LLM-driven approaches remains consistent with our observations on the training problem instances, with BAG demonstrating superior performance.

Table 11: The best normalized (higher is better) AUC achieved by four LLM-driven approaches on the **test instances** of 23 pbo problems. Additionally, the best results obtained by the oracle codes are also considered. Results are aggregated over three LLMs. Each normalized AUC value is followed by its corresponding rank in brackets. The best entries are underlined.

| Function ID | BAG | EoH | LLaMEA | ReEvo |
|:-----------:|:---:|:---:|:------:|:-----:|
| F1 | 0.9851 (2) | 0.9793 (3) | 1.0000 (1) | 0.9640 (4) |
| F2 | 0.8652 (3) | 1.0000 (1) | 0.9476 (2) | 0.8304 (4) |
| F3 | 0.9858 (2) | 0.9759 (3) | 1.0000 (1) | 0.9561 (4) |
| F4 | 0.9965 (2) | 0.9810 (3) | 1.0000 (1) | 0.9739 (4) |
| F5 | 0.9433 (4) | 1.0000 (1) | 0.9547 (3) | 0.9794 (2) |
| F6 | 1.0000 (1) | 0.9674 (3) | 0.9650 (4) | 0.9712 (2) |
| F7 | 1.0000 (1) | 0.9672 (2) | 0.9670 (3) | 0.9488 (4) |
| F8 | 0.9890 (3) | 0.9890 (3) | 1.0000 (1) | 0.9910 (2) |
| F9 | 1.0000 (1) | 0.9818 (3) | 0.9763 (4) | 0.9949 (2) |
| F10 | 1.0000 (1) | 0.9503 (3) | 0.9914 (2) | 0.9432 (4) |
| F11 | 1.0000 (1) | 0.9784 (2) | 0.8996 (4) | 0.9041 (3) |
| F12 | 0.7661 (4) | 1.0000 (1) | 0.8116 (3) | 0.8734 (2) |
| F13 | 0.9670 (3) | 0.9896 (2) | 0.8896 (4) | 1.0000 (1) |
| F14 | 0.6223 (4) | 0.9439 (2) | 1.0000 (1) | 0.7376 (3) |
| F15 | 1.0000 (1) | 0.9035 (2) | 0.8589 (4) | 0.8700 (3) |
| F16 | 1.0000 (1) | 0.8760 (4) | 0.9236 (3) | 0.9277 (2) |
| F17 | 1.0000 (1) | 0.9465 (2) | 0.7563 (4) | 0.8541 (3) |
| F18 | 0.9435 (4) | 1.0000 (1) | 0.9846 (2) | 0.9550 (3) |
| F19 | 0.9881 (2) | 0.9866 (3) | 1.0000 (1) | 0.9501 (4) |
| F20 | 1.0000 (1) | 0.9663 (4) | 0.9944 (2) | 0.9867 (3) |
| F21 | 0.8989 (4) | 1.0000 (1) | 0.9378 (2) | 0.9059 (3) |
| F22 | 1.0000 (1) | 0.9789 (2) | 0.9664 (3) | 0.8736 (4) |
| F23 | 0.9891 (2) | 0.9605 (3) | 0.9247 (4) | 1.0000 (1) |
| Mean | 0.9539 | 0.9705 | 0.9456 | 0.9300 |
| Std | 0.0918 | 0.0309 | 0.0652 | 0.0655 |
| Average Rank | 2.13 | 2.35 | 2.57 | 2.91 |

Table 12: The best normalized (higher is better) AUC achieved by four LLM-driven approaches on the **test instances** of 24 bbob problems. Additionally, the best results obtained by the oracle codes are also considered. Results are aggregated over three LLMs. Each normalized AUC value is followed by its corresponding rank in brackets. The best entries are underlined.

|  | BAG | EoH | LLaMEA | ReEvo |
|---|---|---|---|---|
| F1 | 1.0000 (1) | 0.9981 (2) | 0.9936 (3) | 0.8620 (4) |
| F2 | 1.0000 (1) | 0.9278 (2) | 0.8061 (4) | 0.9265 (3) |
| F3 | 1.0000 (1) | 0.8547 (2) | 0.7105 (3) | 0.5020 (4) |
| F4 | 1.0000 (1) | 0.6185 (3) | 0.9096 (2) | 0.5011 (4) |
| F5 | 1.0000 (1) | 0.9985 (2) | 0.9951 (3) | 0.9947 (4) |
| F6 | 1.0000 (1) | 0.8830 (2) | 0.8376 (4) | 0.8800 (3) |
| F7 | 1.0000 (1) | 0.5242 (4) | 0.6788 (2) | 0.5714 (3) |
| F8 | 0.9249 (2) | 0.6041 (3) | 1.0000 (1) | 0.5849 (4) |
| F9 | 1.0000 (1) | 0.5240 (4) | 0.7333 (3) | 0.7510 (2) |
| F10 | 1.0000 (1) | 0.4936 (4) | 0.6164 (2) | 0.4988 (3) |
| F11 | 1.0000 (1) | 0.5210 (3) | 0.5533 (2) | 0.5117 (4) |
| F12 | 1.0000 (1) | 0.6619 (2) | 0.4602 (4) | 0.6049 (3) |
| F13 | 1.0000 (1) | 0.6091 (2) | 0.4771 (4) | 0.5435 (3) |
| F14 | 1.0000 (1) | 0.8740 (3) | 0.9075 (2) | 0.7139 (4) |
| F15 | 1.0000 (1) | 0.4892 (2) | 0.4574 (4) | 0.4692 (3) |
| F16 | 1.0000 (1) | 0.7936 (2) | 0.6624 (4) | 0.6956 (3) |
| F17 | 0.8763 (3) | 0.8704 (4) | 0.9899 (2) | 1.0000 (1) |
| F18 | 1.0000 (1) | 0.6128 (2) | 0.4848 (4) | 0.5607 (3) |
| F19 | 0.9421 (3) | 0.9425 (2) | 0.8973 (4) | 1.0000 (1) |
| F20 | 1.0000 (1) | 0.6064 (3) | 0.6220 (2) | 0.4402 (4) |
| F21 | 0.3698 (4) | 1.0000 (1) | 0.6400 (3) | 0.9539 (2) |
| F22 | 0.6244 (2) | 0.4633 (3) | 1.0000 (1) | 0.4337 (4) |
| F23 | 1.0000 (1) | 0.7592 (4) | 0.8635 (2) | 0.7973 (3) |
| F24 | 0.2669 (4) | 1.0000 (1) | 0.2964 (3) | 0.3068 (2) |
| Mean | 0.9169 | 0.7346 | 0.7330 | 0.6710 |
| Std | 0.2015 | 0.1920 | 0.2096 | 0.2103 |
| Average Rank | 1.50 | 2.58 | 2.83 | 3.08 |

## K PROPORTION OF FAILED CODE GENERATION

Although LLMs are capable of generating algorithm codes, they may still produce programmes that fail to execute due to syntax errors or other logic issues. In our experiments, we set a maximum 3000s CPU time limit for testing each problem. Note that we conduct 5 independent runs (600s) for each problem. Runs that either cannot be executed correctly or fail to produce any results within this limit are assigned a negative infinity fitness value (the worst possible). Figures 16 and 17 present the average proportions of failed code generation within 100 LLM querying budget for each tested problem. The plots do not reveal a clear difference among the compared LLM-driven optimization methods, as such failures primarily rely on the underlying LLM models rather than the proposed LLM-driven optimization frameworks.

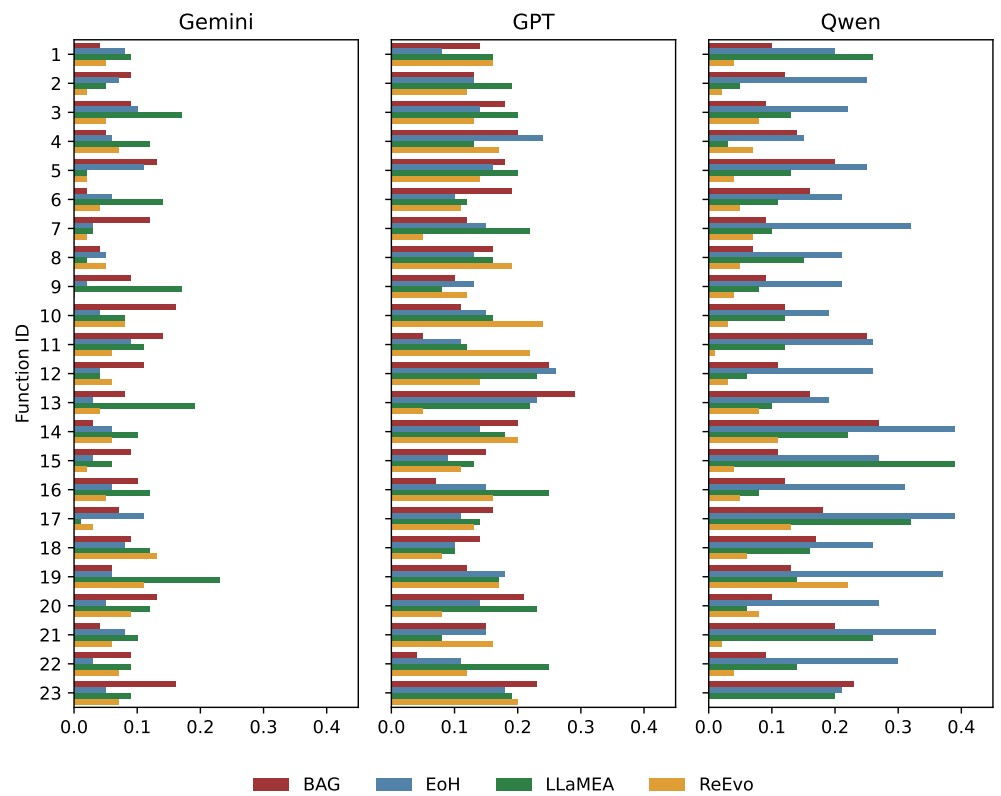

Figure 16: Averaged proportions of failed code generation across 23 pbo problems for the compared LLM-driven approaches. The performances of using Gemin, GPT, and Qwen are plotted from left to right.

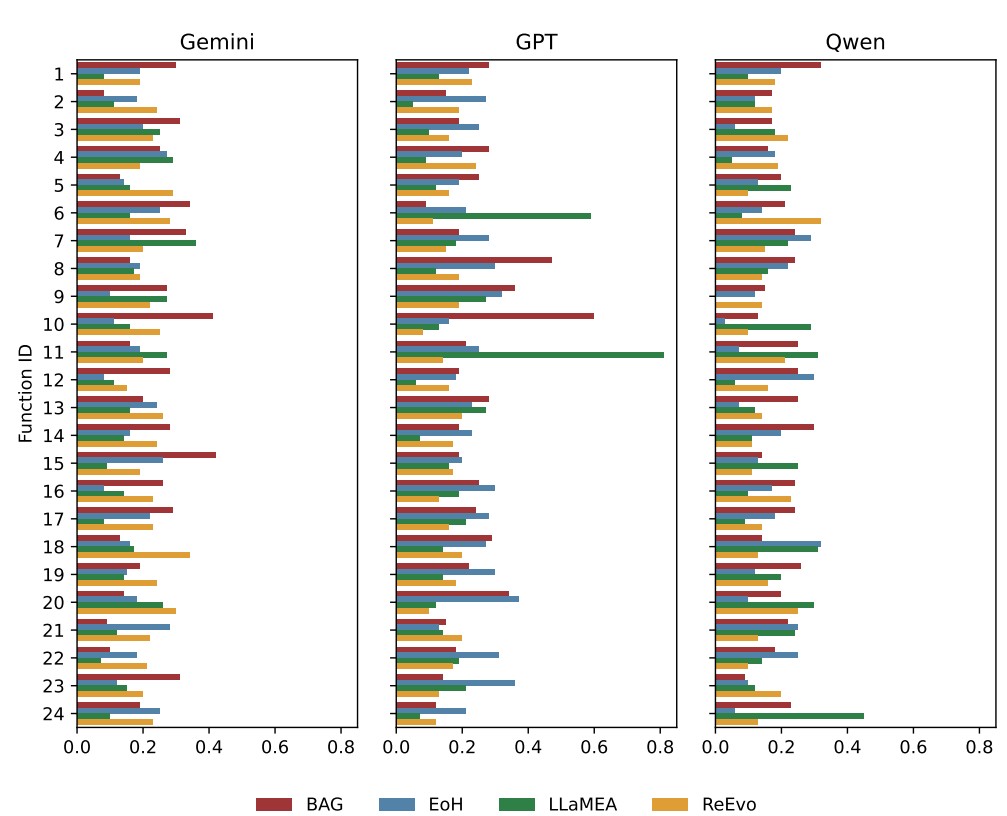

Figure 17: Averaged proportions of failed code generation across 24 bbob problems for the compared LLM-driven approaches. The performances of using Gemin, GPT, and Qwen are plotted from left to right.

