# OpenReview forum: "Automated Algorithm Design with LLMs: A Benchmark-Assisted Approach to Black-Box Optimization"
_ICLR.cc/2026/Conference — Submitted to ICLR 2026_

### Official Review · Reviewer_wS5e · 2025-10-25

**Soundness:** 2
**Presentation:** 2
**Contribution:** 2
**Rating:** 2
**Confidence:** 3

**Summary:**

This paper studies the automated algorithm design based on LLMs for black box optimization. Using token-wise relevance attribution (AttnLRP), the authors find that example code in prompts has a dominant influence on LLM outputs. Building on this, they introduce a benchmark-assisted guided evolutionary approach (BAG), leveraging historical high-quality algorithm codes to guide LLM-driven search. Experiments on two BBO benchmarks (pbo and bdob) show that BAG improves performance and robustness against several LLM-driven methods.

**Strengths:**

* The use of AttnLRP to quantify prompt component impact on output code is interesting and new to optimization tasks.
* The proposed method yields better performance for the two studied benchmark problem sets.
* The appendix provides detailed breakdown results and illustrations of the example code.

**Weaknesses:**

* The proposed concept of "benchmark guided approach" appears to be a rebranding of in-context learning, which is well-established in LLM literature. The distinct contribution beyond standard in-context learning practices remains unclear. The analysis lacks an in-depth discussion.

* While the AttnLRP analysis provides interesting insights that seem to be applicable to broader LLM-driven algorithm design methods across various optimization tasks, the paper narrows its focus to black-box optimization only. The motivation for this restriction is not explained, and it remains unclear whether the proposed method could be applied to other optimization domains.

* Also, the analysis lacks an in-depth discussion. Section 4 stops at showing that "example code has the highest attribution." The robustness of this conclusion is not examined.

* I also have concerns that the approach's effectiveness appears to depend heavily on the quality of exemplar algorithms in the benchmark set. The paper lacks sufficient analysis and discussion regarding whether the proposed method may overfit to the provided benchmark algorithms and how exemplar quality and diversity impact overall performance.

* The evaluation lacks comparison with other in-context learning techniques from the broader LLM literature. Additionally, detailed ablation studies and sensitivity analyses are absent.

**Questions:**

Please address the above concerns.

---

> ### Author Response · Authors · 2025-11-25
>
> We sincerely appreciate the reviewer’s comments. We have revised and added content to address the concerns, and the following are detailed responses.
>
> 1. **LLM-driven Optimization and In-context Learning.** Our work focuses on LLM-driven optimization, meaning we use LLMs to search for better algorithms. This concept fundamentally differs from in-context learning, in which an LLM adapts its behaviour within a single prompt. In this work, our goal is to optimize the search process itself, rather than rely on within-prompt adaptation. We have clarified this in lines 291-296.
> 2. **Motivation and Generalization.**  We have clarified our motivation for focusing on the black-box optimization in the revision (lines 66-71).
>     One key contribution is demonstrating that leveraging benchmark codes can better guide LLMs in the search for better algorithms. Since LLM-driven optimization methods commonly treat the search as a black box process, which is agnostic to the target problem, our approach can naturally be adopted in other domains.
> 3. **Relevance Score Robustness and More Detailed Analysis.**
> The relevance scores in Table 1 are normalized across 100 prompt-code pairs, ensuring robustness of the result. We also provide additional individual analysis results on our open-source GitHub repository.
>
>     In response to the request for deeper analysis, we have added additional analysis of the long-term impact of prompt components using the CodeBLEU method to measure the code similarity among all generated algorithm codes, indicating that the benchmark algorithms we provide significantly contribute to the superior performance of our BAG methods (see page 9).
> 4. **Generation across Problem Instances.** Since our goal is to find better algorithms for a given problem, overfitting to specific instances is not a concern. Still, to address this point, we provide additional validation on unseen problem instances in Appendix J, where the relative performance remains consistent, confirming that BAG generalizes well.
>
>     Note that the pbo and bbob benchmark suites consist of 47 problems with different landscapes. Each benchmark problem can be formulated as various instances, and in our experiments, we evaluate five instances per run. These results support that our conclusions generalize in black-box optimization.
> 5. **Comparison with In-context Learning.** We appreciate the concerns on comparison with in-context learning and our work. We have added a related discussion in lines 291-296. An ablation study on $q$ is added to Appendix F. Additional results on testing unseen problem instances are added to Appendix J.

---

### Official Review · Reviewer_h8xr · 2025-10-28

**Soundness:** 3
**Presentation:** 3
**Contribution:** 2
**Rating:** 4
**Confidence:** 4

**Summary:**

This paper analyses the token-wise relevance of the prompt to the generated black-box optimization algorithm code and finds that the code-related content obtains a strong influence. The authors then propose a benchmark-guided algorithm code generation method, which uses a set of algorithm codes as examples and asks LLMs to refine the examples or generate new algorithms. Experimental results show that the proposed method achieves better performance than some LLM-based optimization code generation baselines.

**Strengths:**

1. This paper analyses the token-wise contributions in the prompts for optimization algorithm code generation, which could be helpful for futher prompt design.

2. The authors introduce more expert knowledge in black-box optimization to enhance the quality of the generated algorithms.

3. Experimental results validate the advantage of the proposed method on some of the existing methods.

**Weaknesses:**

1. Several related works on optimization code generation are not mentioned, such as LLaMoCo [1], which directly generates algorithm code using algorithm benchmarking during training, and LLMOPT [2], which generates operator code via prompting.

2. In the heatmap, the most relevant token appears to be ''numpy'' in the import statement. The LLM also assigns significant attention to unimportant tokens such as ''#'' and '')'', while the performance score of the code is not considered. This suggests the results may be biased, making the LLM-based code search appear almost random and unrelated to the target problem or algorithmic performance.

3. In the experiment, a comparison between the proposed method and the benchmark algorithms (e.g., CMA-ES, DE, PSO) is necessary to validate its effectiveness. Given the computational resources consumed for LLM inference and algorithm evaluation, the generated algorithm is expected to outperform the examples.

[1] Ma, Zeyuan, et al. "Llamoco: Instruction tuning of large language models for optimization code generation." arXiv preprint arXiv:2403.01131 (2024).

[2] Huang, Yuxiao, et al. "Autonomous multi-objective optimization using large language model." IEEE Transactions on Evolutionary Computation (2025).

**Questions:**

1. In the Initialization of the method, the authors use a selected "promising" example code. How is this preferred algorithm chosen? How can users determine which algorithm is "promising" for a target problem without prior knowledge?

2. In Algorithm 1, the method introduce a parameter, frequency factor q. However, the authors do not specify its value or the methodology for setting it.

3. The codes generated by LLMs may not always be correct and may contain syntax errors, runtime errors, or other logical issues. How are these error codes handled in your framework? Does their failure feedback influence the scoring mechanism or inform the LLMs during training?

---

> ### Author Response · Authors · 2025-11-25
>
> We sincerely appreciate the comments and the pointers to related work. We are delighted to learn about these works and to add them to our related work. Since one paper was published this year and the other one is currently on arXiv, we do not include them in the comparison, but we will definitely follow up for future studies.
> The detailed responses to the questions are below.
>
> 1. **Choice and Impact of Benchmark Algorithms.**
> As described in our experimental settings section, we select the Top five algorithms for each pbo benchmark from the corresponding cited work, and five widely adopted algorithms for continuous BBO in bbob. The selected algorithms are established and promising methods, though they are not guaranteed to be globally optimal across all possible algorithm designs.
>
>     If genuinely no useful benchmark algorithms existed, BAG would naturally perform similarly to prior LLM-driven methods. However, prior knowledge, to some extent, always exists in practical applications.
>
> 2. **Choice of q.**
> We appreciate the reviewer pointing out this issue. We have tested the impact of q on BAG performance, and q = 10 is the best choice among the six values (see Appendix F). With this choice and a budget of 100 LLM queries, BAG can, on average, utilize each of the five benchmark algorithms twice in a run, which provides a balanced search process.
>
> 3. **Handling Failed Codes.**
> For the codes that cannot be executed correctly, we denote their fitness by infinity (the worst). This is a common issue across LLM-driven optimization methods, and we now report the frequency of this error in Figures 16 and 17 of Appendix K (pages 40 and 41).

---

### Official Review · Reviewer_Fqie · 2025-10-31

**Soundness:** 3
**Presentation:** 3
**Contribution:** 2
**Rating:** 4
**Confidence:** 4

**Summary:**

This paper presents BAG, a framework for automated algorithm design using LLMs in the setting of black-box optimization. The authors first conduct a token-level attribution study using AttnLRP to identify which parts of prompts most influence algorithm generation, concluding that code examples dominate the contribution. Motivated by this finding, they propose to embed prior benchmark algorithms from the PBO and BBOB suites into the prompts to guide LLMs toward promising search regions. Experiments on 47×5 benchmark instances using three LLMs (Gemini, GPT, and Qwen) show that BAG achieves higher AUC performance than existing LLM-driven optimization frameworks such as EoH, LLaMEA, and ReEvo. The paper claims that integrating benchmark knowledge improves the efficiency, robustness, and interpretability of LLM-based algorithm design.

**Strengths:**

1. The paper conducts large-scale evaluation across two widely accepted benchmark suites (PBO, BBOB) and three different LLMs, ensuring reproducibility and empirical rigor.
2. Incorporating AttnLRP to analyze prompt token relevance is a useful attempt to make LLM-driven algorithm generation more interpretable, identifying which parts of a prompt truly influence generated code.
3. The paper is well-organized and clearly written, with figures and tables that effectively communicate experimental results. The released code repository also enhances transparency.

**Weaknesses:**

1. The proposed BAG framework primarily reuses existing benchmark algorithms as prompt examples. While this improves performance, it effectively narrows the search to regions of known good solutions rather than discovering new algorithms. Thus, the method relies heavily on prior knowledge injection rather than genuine algorithmic innovation, which weakens the originality of the contribution.
2. The paper frames BAG as a mechanism for discovering new optimization algorithms, but since benchmark algorithms are directly embedded into prompts, the system is refining known strategies rather than inventing novel ones. The contribution fits better under knowledge-guided prompt design than under automated algorithm design.
3. The AttnLRP analysis merely confirms an intuitive result—that code tokens dominate influence in code generation. This offers limited new understanding of LLM behavior beyond what prior intuition already suggested.
4. No ablation is provided for the frequency factor q, the size of the benchmark set, or the quality of the prior examples. It remains unclear whether BAG still performs well if benchmark algorithms are noisy or suboptimal. Without this, claims about robustness are not substantiated.
5. The core idea—guiding LLMs via benchmark priors—feels incremental relative to recent literature (e.g., EoH, ReEvo, LLaMEA). The results, while positive, are expected given the strong prior information injected into the prompts."

**Questions:**

1. How does BAG perform if benchmark algorithms are replaced with suboptimal or randomly perturbed versions?
2. Can BAG generate genuinely new algorithmic structures beyond minor code variations of the provided benchmarks?
3. Why not formalize BAG as a knowledge-transfer or fine-tuning paradigm instead of positioning it as algorithm discovery?
4. Is there evidence that BAG generalizes beyond PBO/BBOB, or does it overfit to those benchmarks?"

---

> ### Author Response · Authors · 2025-11-25
>
> We sincerely appreciate the helpful comments and questions. We have revised the paper addressing the questions. The following are some detailed responses.
>
> 1. **Impact of the Benchmark Algorithm.**
> In our experimental setting, the selected algorithms are promising ones within the known benchmark studies, but they are not guaranteed to be optimal if considering all possible algorithm designs. Nonetheless, our results demonstrate the superiority of BAG. Regarding the question about providing the 'worst' benchmark algorithm, BAG will not collapse even in this extreme case. With the (1+1) strategy, BAG will immediately discard the ‘worst’ algorithm when a new algorithm is created. After this, BAG will perform similarly to the compared methods.
> 2. **Potential novel algorithms.**
> We have included additional analysis of the relevance of the provided codes and the generated algorithm on pages 9 and 10. On the one hand, we expect BAG to produce better algorithmic code by revising the provided code. Meanwhile, the BAG method enables the development of novel algorithms, as shown in line 14 of Algorithm 1. So the answer is 'Yes' to the question.
> 3. **Algorithm Discovery.**
> The BAG method essentially applies the (1+1) evolutionary algorithm to search for better algorithms, and its search process is controlled by utilizing benchmark algorithms. Therefore, we are dealing with an optimization problem of finding the optimal algorithm, and we think it is more appropriate to address it as algorithm discovery. Related explanation has been added on page 6.
> 4. **Generalization.**
> The pbo and bbob benchmark suites consist of 47 problems with different landscapes. Each benchmark problem can be formulated as various instances, and in our experiments, we evaluate five instances per run.
>
>      Since our goal is to find better algorithms for a given problem, overfitting to specific instances is not a concern. Still, to address this point, we provide additional validation on unseen problem instances in Appendix J, where the relative performance remains consistent, confirming that BAG generalizes well.

---

### Official Review · Reviewer_eyA5 · 2025-11-01

**Soundness:** 2
**Presentation:** 2
**Contribution:** 2
**Rating:** 2
**Confidence:** 5

**Summary:**

This paper proposes a framework named the "Benchmark-assisted Guided evolutionary Approach" (BAG) for automated algorithm design in black-box optimization  using Large Language Models (LLMs). The core idea is to leverage a pre-existing set of high-performance benchmark algorithms to guide the evolutionary process of the LLM. The authors first use attribution analysis (AttnLRP) to argue for the importance of code examples in the generated output of LLMs. They then validate the BAG framework on the pbo and bbob BBO benchmark suites, claiming superior performance over several other LLM-based methods like EoH and ReEvo. The paper's main conclusion is that integrating domain knowledge from benchmarks can effectively enhance the performance of LLM-driven algorithm design.

**Strengths:**

The paper provides experimental evidence for the common intuition that code examples are crucial in prompt engineering by quantitatively analyzing prompt components using AttnLRP.

**Weaknesses:**

1. The core idea of this paper has significant overlap with a recent series of LLM-driven evolutionary algorithms, particularly EoH and ReEvo. These prior works have already established the basic paradigm of using an LLM as an evolutionary operator (e.g., for mutation or crossover) to iteratively generate and improve algorithmic code. The BAG framework is essentially a minor adjustment to this paradigm, introducing a strategy to "sample from a benchmark pool and improve." This is not a fundamental innovation but rather a simple heuristic sampling strategy added to an existing framework. The "guidance" effect claimed by the paper has already been embodied in EoH's "evolution of thoughts" and ReEvo's "reflective evolution"; BAG merely replaces the guidance source from LLM-generated "thoughts" to external "high-performing algorithms."
2. This is arguably the most critical flaw of the study. The experimental evaluation is entirely confined to comparisons with other LLM-driven methods, ignoring the highly optimized non-LLM methods developed in the field of Automated Algorithm Design (AAD). To substantiate claims of superiority, the method could be benchmarked against established, powerful domain-specific baselines, such as advanced automated algorithm evolution frameworks (e.g., OpenEvolve) or efficient surrogate-based Bayesian Optimization methods.
3. The empirical evaluation is restricted to the pbo and bbob benchmark suites. While these are classic benchmarks in BBO, they primarily represent unconstrained numerical optimization problems. The scope of AAD is far broader, encompassing more diverse and complex domains such as combinatorial optimization (e.g., TSP, scheduling) and constrained optimization. Confining the evaluation to two similar testbeds significantly limits the generalizability of the conclusions. The study fails to demonstrate whether BAG remains effective in more diverse and challenging problem domains.

**Questions:**

1. Your experimental evaluation only includes other LLM-driven methods. Why did you not compare BAG against established, state-of-the-art non-LLM frameworks from the AAD field (such as OpenEvolve) or powerful general-purpose BBO solvers (like Bayesian Optimization)?
2. The core contribution of the paper appears to be the demonstration that "using high-quality algorithms as examples can better guide an LLM." This conclusion is not surprising and could even be considered intuitive. Given the high similarity of the BAG's evolutionary loop to that of EoH and ReEvo, could you precisely articulate what the fundamental methodological differences are between BAG and these prior works, beyond the strategy of "selecting parents from an external benchmark pool"?
3. The success of the BAG framework seems highly dependent on a readily available, high-quality set of benchmark algorithms (Abench). For emerging problem domains that lack a mature benchmark community and a set of recognized high-performing algorithms, would the BAG method fail entirely? To what extent are the scalability and generalizability of this method limited by its strong reliance on prior knowledge?

---

> ### Author Response · Authors · 2025-11-25
>
> We sincerely appreciate your comments and questions. We note that the reviewer is especially interested in the performance and setup of our BAG method, and we are delighted to address the details below. We hope the general reply can clarify our motivation and the value of this work.
>
> The following are the responses to the questions:
> 1. **The Choice of Baselines.**
> - This work investigates the impact of prompt design in LLM-driven optimization methods and aims to improve such methods. Therefore, we choose to compare with only LLM-driven methods. The three baselines EoH, LLaMEA, and ReEvo are the most recently published packages, representing the state of the art.
> - Regarding OpenEvolve and AlphaEvolve, these two packages were released within 4 months of our paper submission. According to ICLR policy, authors should not be expected to include comparisons with such works, so we believe it would be inappropriate to require such comparisons.
> - As for Bayesian Optimization, although widely used for algorithm configuration, it is not an LLM-driven technique and operates on tuning parameters instead of code. Applying BO requires first selecting an algorithm for the given problem and determining parameters to be tuned. This is already challenging for the black-box optimization. Therefore, we consider BO falls out of the scope of meaningful comparisons.
>
> 2. **Novel Findings and Fair Evaluation.**
> - One of our key conclusions is ‘using benchmark algorithms can better guide LLMs’. While this may appear intuitive, existing methods have not validated this behavior. In contrast, they still put effort into designing prompts using linguistic descriptions. This work is the first to conduct a comprehensive analysis of prompt-component contributions in LLM-driven optimization.
> - We have explained our evolutionary search procedure in lines 337-341. We adopted a comparable evolutionary search setup to ensure fair evaluation without introducing complex strategies. The (1+1) EA is theoretically and empirically recognized as a safe and competitive strategy when the landscape of the search space is known, making it an appropriate choice for validating the effectiveness of our prompt design.
>
> 3. **Utilization of Benchmark Algorithms.**
> - By providing a set of benchmark problems, we can guarantee that the BAG framework can achieve promising results. Without such guidance, our method will intuitively perform similarly to the compared methods. However, prior knowledge, to some extent, always exists in practical applications. We would also like to mention that additional analysis of cooperating benchmark algorithms is added on page 9.

---

### Author Response · Authors · 2025-11-25
**General Reply to All Reviewers**

We sincerely appreciate all reviewers’ comments and questions. Also, we are sorry for the relatively late response, as we spent time conducting additional experiments.

In this general reply, we would like to address the key concepts that reviewers have misunderstood or overlooked.

1. **Core Contribution beyond Performance.**
Overall, we observe that all reviewers focus on the setup and performance of our proposed BAG methods while relatively ignoring another important contribution of our work: the analysis of the contribution of prompt components, which can provide valuable guidelines for future LLM-driven optimization tasks.

      While researchers aim to promote LLMs to solve optimization tasks across various applications, we are **among the first (and perhaps the only)** initiatives to systematically investigate efficient prompt design for LLM-driven optimization, which can inform more advanced evolutionary search design in the future. Although some reviewing opinion thinks that our finding ‘provides example code to obtain the most significant impact on the output’ is intuitive, we would like to respectfully emphasize that such a fact requires many experiments and analysis to validate. Our work indicates a better way to control LLMs to generate novel algorithms by providing diverse benchmark codes rather than simply querying ‘create a new algorithm’.

2. **Design for Fair Comparison.**
Based on the AttnLRP analysis results, we propose the BAG method, using the simple yet effective (1+1) EA search strategy, to examine the use of benchmark algorithms to guide LLMs. Detailed explanations of this setting can be found in lines 337-341. We would like to highlight that we intentionally avoid more complex evolutionary strategies to fairly examine our idea of using benchmark algorithms.

3. **Additional Experiments are added.**
Additional experiments have been conducted to address reviewers' concerns. Specifically, we have included additional comparisons on unseen test problem instances. Also, we provide additional analysis on the effectiveness of our method using the CoreBLEU metric.

We address this general reply here to highlight our contribution to analyzing the contribution of prompt components in LLM-driven optimization. Also, additional experiments have been added in the revision to better validate our contributions. We have submitted a revision in which modifications have been highlighted in Blue. More detailed responses are followed in individual comments. We look forward to discussing further with the reviewers.

---

### Meta-Review · Area_Chair_ZGjv · 2026-01-06

**Summary:**

This submission proposes a benchmark-assisted guided evolutionary approach (BAG) for LLM-driven black-box optimization, with the core insight that high-quality benchmark code examples enhance prompt effectiveness. While the AttnLRP-based analysis of prompt components and supplementary experiments (e.g., unseen instance validation) demonstrate the authors’ efforts to address reviewer concerns, the work falls short of meeting ICLR’s standards for novelty and rigor.
Key concerns raised by reviewers remain unaddressed sufficiently: First, the core contribution overlaps significantly with existing LLM-driven evolutionary methods (e.g., EoH, ReEvo), as BAG primarily replaces internal "thought/reflective" guidance with external benchmark algorithms—an incremental adjustment rather than fundamental innovation. Second, the evaluation is overly narrow: confining comparisons to LLM-driven baselines ignores state-of-the-art non-LLM automated algorithm design frameworks (e.g., OpenEvolve), and limiting tests to PBO/BBOB (unconstrained numerical optimization) undermines generalizability to broader domains like combinatorial optimization. Third, the method’s heavy reliance on high-quality benchmark pools raises critical scalability questions for emerging problem domains, which the authors’ rebuttal (citing "inherent prior knowledge") does not adequately resolve. Additionally, the AttnLRP analysis confirms intuitive findings about code example importance without providing deeper insights into LLM behavior, and ablation studies for key parameters (e.g., frequency factor q) and benchmark quality/diversity are insufficient to substantiate robustness claims.
While the prompt design analysis offers modest value for future LLM-driven optimization research, the work’s incremental nature, limited evaluation scope, and unresolved foundational concerns prevent acceptance. We encourage the authors to expand the evaluation to non-LLM baselines, explore broader application domains, and deepen the analysis of benchmark dependency to strengthen the contribution in future submissions.

**Reviewer Concerns:**

Novelty & Incrementality:
Overlap with prior works (EoH/ReEvo) – rebuttal frames it as "first systematic prompt component analysis" but fails to demonstrate fundamental methodological innovation beyond swapping guidance sources.
Misalignment between positioning (algorithm discovery) and practice (refining known benchmarks) (Reviewer Fqie’s concern on knowledge-transfer vs. discovery remains unresolved).

Evaluation Limitations:
Lack of comparison with state-of-the-art non-LLM AAD frameworks (e.g., OpenEvolve) – rebuttal’s policy-based justification does not address the core concern of validating superiority against domain standards (Reviewer eyA5).
Restriction to PBO/BBob (unconstrained numerical optimization) – no evidence of generalization to combinatorial/constrained optimization (Reviewers eyA5, wS5e).

Analysis Gaps:
Insufficient ablation studies (benchmark set size, quality/diversity impact; Reviewers Fqie, wS5e).
Bias in AttnLRP results (overattention to trivial tokens like "numpy"/"#" without addressing performance correlation; Reviewer h8xr).

Scalability:
Reliance on high-quality benchmark pools – no solution for emerging domains with no mature benchmarks (Reviewer eyA5).

Unaddressed Comparisons:
Lack of contrast with other in-context learning techniques (Reviewer wS5e).

**Reviewer Scores:**

N/A

---

### Decision · Program_Chairs · 2026-01-26

Reject